# The mechanism of error induction by the antibiotic viomycin provides insight into the fidelity mechanism of translation

**Mikael Holm, Chandra Sekhar Mandava, Måns Ehrenberg, Suparna Sanyal\***

Department of Cell and Molecular Biology, Uppsala University, Uppsala, Sweden

**Abstract** Applying pre-steady state kinetics to an *Escherichia-coli*-based reconstituted translation system, we have studied how the antibiotic viomycin affects the accuracy of genetic code reading. We find that viomycin binds to translating ribosomes associated with a ternary complex (TC) consisting of elongation factor Tu (EF-Tu), aminoacyl tRNA and GTP, and locks the otherwise dynamically flipping monitoring bases A1492 and A1493 into their active conformation. This effectively prevents dissociation of near- and non-cognate TCs from the ribosome, thereby enhancing errors in initial selection. Moreover, viomycin shuts down proofreading-based error correction. Our results imply a mechanism in which the accuracy of initial selection is achieved by larger backward rate constants toward TC dissociation rather than by a smaller rate constant for GTP hydrolysis for near- and non-cognate TCs. Additionally, our results demonstrate that translocation inhibition, rather than error induction, is the major cause of cell growth inhibition by viomycin.

DOI: https://doi.org/10.7554/eLife.46124.001

## Introduction

Viomycin is the first discovered member of the tuberactinomycin class of bacterial protein synthesis inhibiting antibiotics (*Ehrlich et al., 1951*; *Finlay et al., 1951*), commonly used to treat infections by *Mycobacterium tuberculosis* strains resistant to first-line drugs (*WHO, 2010*). It is a cyclic pentapeptide, that is naturally synthesized by a non-ribosomal peptidyl transferase (*Thomas et al., 2003*). Viomycin impairs the fidelity of tRNA selection (*Marrero et al., 1980*) and reduces the rate of mRNA translocation (*Holm et al., 2016*; *Modolell and Vázquez, 1977*) during the elongation cycle of bacterial protein synthesis. We have recently described the kinetic mechanism by which viomycin inhibits translocation (*Holm et al., 2016*), and here we report on the kinetic mechanism by which viomycin impairs the accuracy of AA-tRNA selection.

During genetic code translation, aminoacyl-transfer RNAs (AA-tRNAs) are delivered to the A site of the ribosome in ternary complex (TC) with elongation factor Tu (EF-Tu) and GTP. For fast and accurate protein synthesis, the ribosome must select for GTP hydrolysis those TCs which contain an AA-tRNA with a base triplet (anticodon) cognate to the base triplet on the mRNA (codon) displayed in the ribosomal A site. Those cognate AA-tRNAs will then be selected for A-site accommodation and peptidyl transfer. In quantitative terms, this means that cognate codon selection of AA-tRNAs for GTP hydrolysis and peptidyl transfer must be characterized by much higher catalytic efficiency ($k_{cat}/K_M$) than near-cognate codon selection. In enzyme kinetics, $k_{cat}$ corresponds to the maximal rate of product formation at saturating substrate concentration and the Michaelis-Menten constant to the substrate concentration at which product formation rate is half maximal. The accuracy by which the ribosome discriminates against a given codon•anticodon mismatch is defined as the ratio between the $k_{cat}/K_M$ values of the cognate and the non-cognate reaction (*Fersht, 1998*).

**\*For correspondence:**
suparna.sanyal@icm.uu.se

**Competing interests:** The authors declare that no competing interests exist.

Selection of AA-tRNA by the ribosome occurs in two phases: initial codon selection before GTP hydrolysis on EF-Tu and proofreading selection after GTP hydrolysis (*Ruusala et al., 1982*; *Thompson and Stone, 1977*). Initial selection begins by TC binding to the ribosomal A/T site, from which TC is either rejected by dissociation from the ribosome or accepted by the triggering of GTP hydrolysis on EF-Tu. The accuracy of initial codon selection is amplified by A-minor interactions between the codon-anticodon helix and the 16S ribosomal RNA (rRNA) monitoring bases A1492, A1493 and G530 (*E. coli* numbering) of the decoding center (*Carter et al., 2000*). It has been suggested that the monitoring bases only flip out from their binding sites in helix 44 (h44) of 16S rRNA to form hydrogen bonds with cognate but not near- or non-cognate codon-anticodon helices (*Carter et al., 2000*). However, in recent crystal structures of A-site accommodated cognate and near-cognate tRNAs the monitoring bases were observed in virtually identical, flipped-out, conformations in all cases (*Demeshkina et al., 2012*; *Demeshkina et al., 2013*). Further, recent cryo-EM structures (*Fislage et al., 2018*; *Loveland et al., 2017*) show that during initial selection the decoding center builds up in an identical step-wise fashion for both cognate and near-cognate tRNAs to a common final state in which all three monitoring bases are in their activated conformations. These structural data agree with kinetics data on initial TC-selection in the absence and presence of aminoglycosides (*Zhang et al., 2018*), suggesting that realistic modeling of initial codon selection requires at least four ribosomal states (*Fislage et al., 2018*; *Loveland et al., 2017*; *Pavlov and Ehrenberg, 2018*; *Zhang et al., 2018*).

In crystal (*Pulk and Cate, 2013*; *Stanley et al., 2010*) and cryo-EM (*Brilot et al., 2013*) structures of the viomycin-bound ribosome, A1492 and A1493 are seen in their active, flipped-out conformation, and viomycin is bound to a site which is occluded by the monitoring bases in their inactive, flipped-in conformation (*Schuwirth et al., 2005*). From these structures it seems likely that association of viomycin to the ribosome requires bases A1492 and A1493 in their flipped-out conformation and that the presence of viomycin on the ribosome will effectively block the monitoring bases from returning from their inactive, flipped-in, conformation. This interplay between viomycin and the monitoring A1492 and A1493 bases could then potentially drive activation of the third monitoring base, G530, thereby triggering GTP hydrolysis in the TC (*Loveland et al., 2017*). Such a conformational-selection mode of viomycin binding agrees well with our previous result that A-site-bound tRNA greatly increases the affinity of viomycin for the ribosome (*Holm et al., 2016*).

Here, we have examined how viomycin reduces the accuracy of tRNA selection on the mRNA translating ribosome. For this, we applied pre-steady state kinetics and mean time analysis to a cell-free protein synthesis system, reconstituted from *E. coli* components of high purity and in vivo like kinetic properties (*Borg et al., 2015*; *Borg and Ehrenberg, 2015*; *Indrisiunaite et al., 2015*; *Johansson et al., 2008*; *Mandava et al., 2012*). Our results are summarized by a kinetic model, which illustrates the mechanism of error induction by viomycin and other drugs in the tuberactinomycin class of antibiotics. We suggest that high accuracy of initial codon selection by cognate TCs is mainly achieved by much smaller backward rate constants toward dissociation of cognate than near-cognate TCs. Our data do not support the previous suggestion that cognate TCs have much larger rate constant for GTP hydrolysis than near-cognate TCs (*Gromadski and Rodnina, 2004*; *Pape, 1999*). We compare the modes of action of aminoglycosides and viomycin by highlighting their functional similarities and differences and use simple modeling techniques to estimate the frequency and distribution of viomycin-induced translational errors in the living cell. With support from the present data, we propose that translocation inhibition, rather than error induction, is the major cause of cell growth inhibition by viomycin.

## Results

### Viomycin acts during initial codon selection of aminoacyl-tRNAs on the ribosome

To study the impact of viomycin on translational accuracy, we designed experiments to measure its effect on the kinetic efficiency ($k_{cat}/K_M$) of GTP hydrolysis by EF-Tu and peptide bond formation for both cognate and near-cognate codon-anticodon interactions. A reaction mixture containing varying concentrations of viomycin and $\mathrm{Phe-tRNA_{GAA}^{Phe}}$ in TC with EF-Tu·GTP was rapidly mixed in a quench-flow instrument with initiated 70S ribosomes displaying either cognate (UUC) or near-

cognate (CUC) codons in the A site. For studying GTP hydrolysis, the TCs contained [3H]GTP and the 70S ribosomes had non-radioactive fMet-tRNA$^{fMet}$ in the P site, while for studying peptide bond formation the TCs contained non-radioactive GTP and the 70S ribosomes had f[3H]Met-tRNA$^{fMet}$ bound in the P site. The reactions were stopped at different times by addition of formic acid and the relative amounts of the reaction products were analyzed by ion-exchange chromatography ([3H] GDP) or RP-HPLC (f[3H]Met-Phe) with on-line radiation detection.

The $k_{cat}/K_M$ for cognate GTP hydrolysis (blue traces in **Figure 1A**) did not change with increasing viomycin concentration from 0 to 1 mM and its average was estimated as 41 ± 0.6 µM$^{-1}$ s$^{-1}$ (**Figure 1B**). In sharp contrast, the $k_{cat}/K_M$ of GTP hydrolysis for the near-cognate reaction (Red traces in **Figure 1A**) increased dramatically with increasing viomycin concentration (**Figure 1B**) from 0.053 ± 0.005 µM$^{-1}$ s$^{-1}$ in the absence of viomycin to 9.2 ± 0.7 µM$^{-1}$ s$^{-1}$ in the presence of 1 mM viomycin, corresponding to a 170-fold reduction in the accuracy of initial codon selection. Viomycin

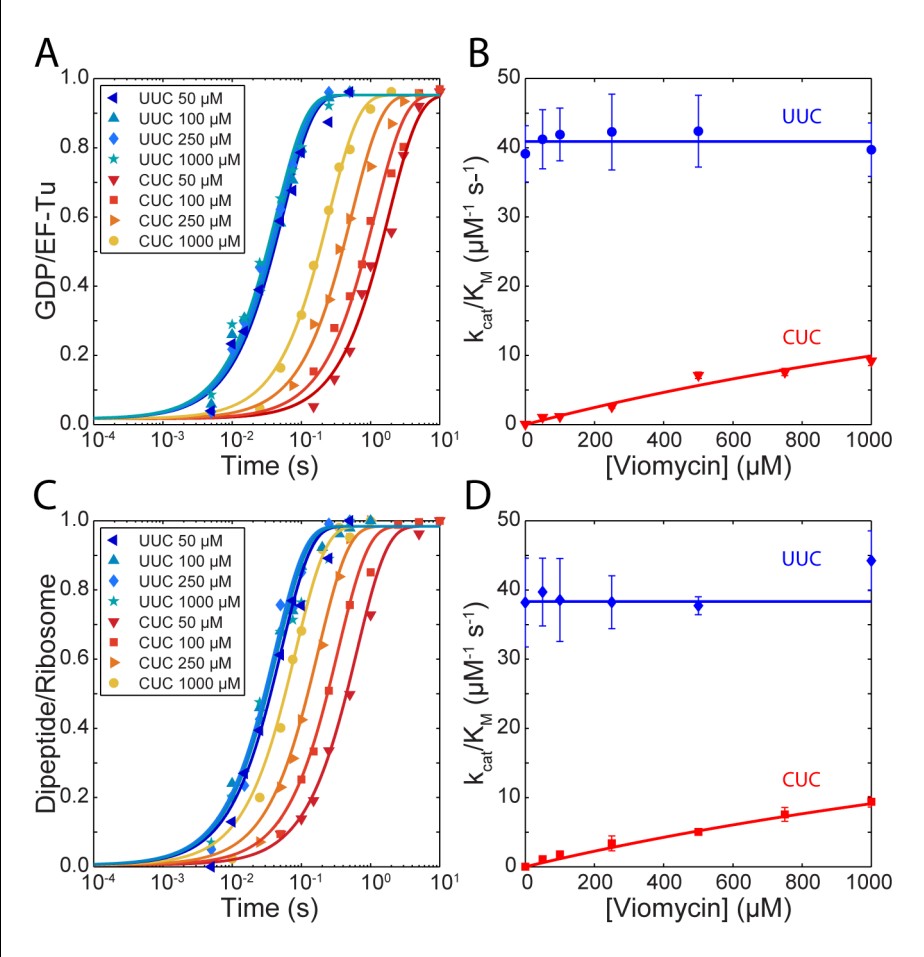

**Figure 1.** Kinetics of tRNA selection in the presence of viomycin. (A) Time courses of [3H] GDP formation by Phe-tRNA$^{Phe}$ containing EF-Tu TCs reacting with 0.5 µM 70S ribosomes with fMet-tRNA$^{fMet}$ in the P site, displaying either a cognate UUC or near-cognate CUC codon in the A site, at the indicated viomycin concentrations. Solid lines represent fits of single exponential equations to the data. (B) of GTP hydrolysis by Phe-tRNA$^{Phe}$ containing TCs estimated from experiments such as those in (A). Solid lines represent either a fit of a constant value (UUC) or **Equation 3** (CUC) to the data. (C) Time courses of f[3H] Met-Phe dipeptide formation for 1 µM Phe-tRNA$^{Phe}$ containing EF-Tu TCs reacting with 70S ribosomes with f[3H]Met-tRNA$^{fMet}$ in the P site, displaying either a cognate UUC or near-cognate CUC codon in the A site, at the indicated viomycin concentrations. Solid lines represent fits of single exponential equations to the data. (D) of dipeptide formation by Phe-tRNA$^{Phe}$ containing EF-Tu TCs estimated from experiments such as those in (C). Solid lines represent either a fit of a constant value (UUC) or **Equation 3** (CUC) to the data. All error bars represent SEM.

DOI: https://doi.org/10.7554/eLife.46124.002

is known to bind to ribosomes with a cognate codon·anticodon interaction in the A site (*Brilot et al., 2013*; *Feldman et al., 2010*; *Holm et al., 2016*; *Stanley et al., 2010*; *Zhou et al., 2012*) and to strongly stabilize such complexes (*Peske et al., 2004*). Hence, these results imply that during initial codon selection viomycin acts on a ribosomal state from which near-cognate, but not cognate, substrates are likely to be rejected in the absence of the drug.

Similar to GTP hydrolysis the $k_{cat}/K_M$ of dipeptide formation with a cognate UUC codon in the ribosomal A site (blue traces in *Figure 1C*) did not change with the addition of viomycin and its average was estimated as $38.3 \pm 0.7 \ \mu M^{-1} \ s^{-1}$ (*Figure 1D*). As with GTP hydrolysis, the $k_{cat}/K_M$ of dipeptide formation with a near-cognate CUC codon in the ribosomal A site (red traces in *Figure 1C*) increased dramatically with increasing viomycin concentration (*Figure 1D*), from $0.0005 \pm 0.00004$ $\mu M^{-1} \ s^{-1}$ in the absence of viomycin to $9.4 \pm 0.8 \ \mu M^{-1} \ s^{-1}$ at 1 mM viomycin. This corresponds to a 21,000 fold reduction in total accuracy from $83,500 \pm 8000$ to $4.0 \pm 0.3$.

The large difference between the $k_{cat}/K_M$ value for near-cognate dipeptide formation ($0.0005 \pm 0.00004 \ \mu M^{-1} \ s^{-1}$) and near-cognate GTP hydrolysis ($0.053 \pm 0.005 \ \mu M^{-1} \ s^{-1}$) in the absence of viomycin is due to proofreading selection. The ratio of these two $k_{cat}/K_M$ values estimates the accuracy of proofreading selection as $115 \pm 15$ (*Zhang et al., 2016*). In contrast, at all tested viomycin concentrations the $k_{cat}/K_M$ values of near cognate GTP hydrolysis and dipeptide formation were virtually identical (*Figure 1B and D*). This means that viomycin-bound ribosomes are incapable of performing proofreading selection; all near- and non-cognate tRNAs that 'survive' initial selection go on to form peptide bonds. Furthermore, even at very low drug concentration almost all near-cognate tRNAs that pass initial selection do so due to the presence of viomycin.

## Viomycin stabilizes a GTPase-deficient TC in contact with both cognate and near-cognate codons on the ribosome

Viomycin is known to strongly stabilize peptidyl-tRNA in the ribosomal A site (*Holm et al., 2016*; *Peske et al., 2004*). To address whether viomycin also stabilizes TCs in the A site during initial codon selection, we estimated the rate of dissociation of both cognate and near-cognate tRNAs in TC (TC$^{H84A}$) with a GTPase-deficient mutant of EF-Tu (EF-Tu$^{H84A}$). In this EF-Tu mutant, an essential histidine in the G-domain has been replaced by alanine (*Daviter et al., 2003*), but formation of TC$^{H84A}$ is unhindered and the mutant TC carries out all partial reactions during initial codon selection, excluding GTP hydrolysis (*Daviter et al., 2003*; *Gromadski and Rodnina, 2004*). Initiated 70S ribosomes with f[$^3$H]Met-tRNA$^{fMet}$ in the P site and a cognate (UUC) or near-cognate (CUC) codon in the A site were equilibrated with TC$^{H84A}$ containing $\mathrm{Phe-tRNA_{GAA}^{Phe}}$ and GTP in the presence of varying concentrations of viomycin. TC$^{H84A}$s were chased from the A site by addition of GTPase proficient TC, containing WT EF-Tu (EF-Tu$^{WT}$) and either $\mathrm{Phe-tRNA_{GAA}^{Phe}}$ or $\mathrm{Leu-tRNA_{GAG}^{Leu2}}$, whichever was cognate for the codon in the A site. The dissociation rate for TC$^{H84A}$s from the A site, defined as the inverse of the average dissociation time, was then estimated from the rate of f[$^3$H]Met-Phe or f[$^3$H]Met-Leu formation (supplementary methods).

The rate of TC$^{H84A}$ dissociation from ribosomes displaying the cognate UUC codon (*Figure 2A*) was $1.21 \pm 0.088 \ s^{-1}$ in the absence of viomycin and decreased from $0.616 \pm 0.044 \ s^{-1}$ in the presence of 1 μM viomycin to $0.198 \pm 0.0275 \ s^{-1}$ at 10 μM viomycin. The corresponding viomycin-induced increase in dissociation mean time ($\tau_{diss}$) is shown in *Figure 2B*. In comparison, dissociation of $\mathrm{Phe-tRNA_{GAA}^{Phe}}$- containing TC$^{H84A}$ from ribosomes displaying the near-cognate CUC codon (*Figure 2C*) was too fast to be estimated using manual mixing techniques in the absence of viomycin, consistent with previous reports (*Gromadski and Rodnina, 2004*; *Johansson et al., 2008*; *Pape, 1999*). In the presence of 50 μM viomycin, the apparent near-cognate dissociation rate was $0.264 \pm 0.055 \ s^{-1}$ and decreased modestly to $0.209 \pm 0.066 \ s^{-1}$ at 200 μM viomycin.

Even in the absence of viomycin, dissociation of cognate TC is much slower (*Figure 2A*) than the forward rate constant of GTP hydrolysis (*Figure 1A*). This indicates that the frequency of cognate tRNA rejection from the state probed by these experiments is very small. However, the fact that cognate TCs are frequently rejected by the ribosome (*Geggier et al., 2010*; *Johansson et al., 2012*; *Zhang et al., 2015*; *Zhang et al., 2016*) suggests the existence of an early initial binding state from which TC rapidly dissociates, as suggested previously (*Geggier et al., 2010*; *Gromadski and Rodnina, 2004*; *Pape et al., 1998*). Together, the decrease in the cognate dissociation rate with increasing viomycin concentration (*Figure 2B*) and the lack of an effect of viomycin on the cognate kinetic

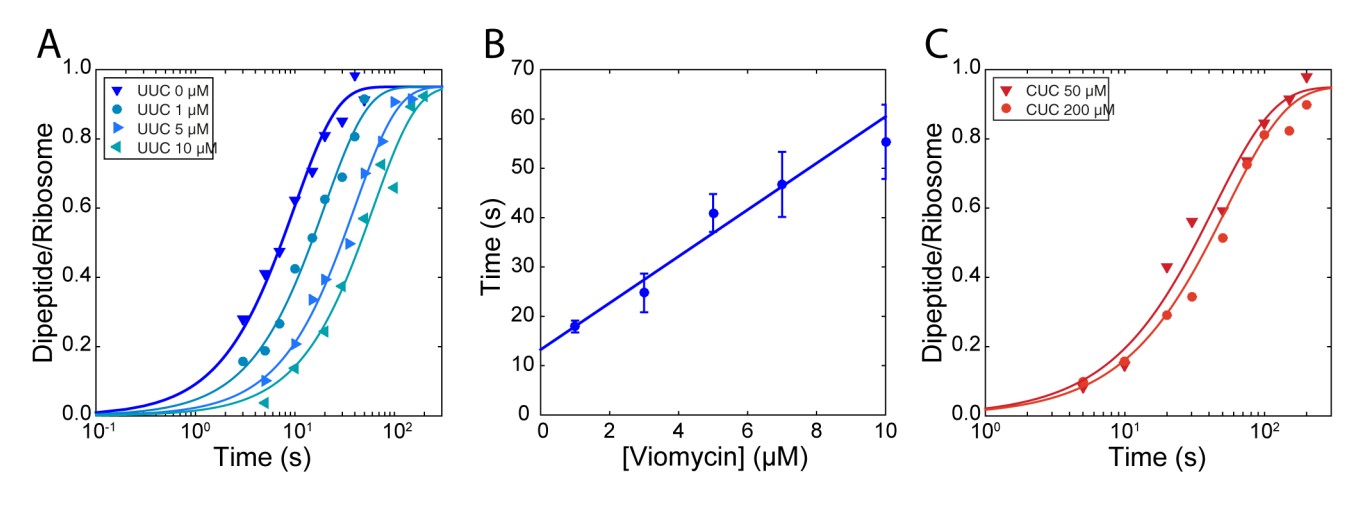

**Figure 2.** Stabilization of ternary complex on the ribosome by viomycin on both cognate and near-cognate codons. (**A**) Time courses of chase experiments were f[3H]Met-Phe dipeptide is formed after chasing of 5 µM containing EF-Tu$^{H84A}$ TCs from 70S ribosomes with f[3H] Met-tRNA$^{fMet}$ in the P site, displaying a cognate UUC codon in the A site, by 0.5 µM containing EF-Tu$^{WT}$ TCs at the indicated viomycin concentrations. Solid lines represent fits of single exponential equations to the data. (**B**) Mean times of f[3H] Met-Phe dipeptide formation reflecting the mean times of containing EF-Tu$^{H84A}$ TC dissociation estimated from experiments such as those in (**A**). Error bars represent SEM. The solid line represents a linear fit to the data. (**C**) Time courses of chase experiments where f[3H]Met-Leu dipeptide is formed after chasing of 5 µM Phe-tRNA$^{Phe}_{GAA}$ containing EF-Tu$^{H84A}$ TCs from 70S ribosomes with f[3H] Met-tRNA$^{fMet}$ in the P site, displaying a near-cognate CUC codon in the A site, by 0.5 µM containing EF-Tu$^{WT}$ TCs at the indicated viomycin concentrations. Solid lines represent fits of single exponential equations to the data.

DOI: https://doi.org/10.7554/eLife.46124.003

efficiency ($k_{cat}/K_m$) observed above (*Figure 1B and D*) implies that viomycin does not stabilize this initial binding state. The drug might, however, further stabilize a late binding state from which cognate TC continues to GTP hydrolysis with high probability whether or not the ribosome is viomycin bound. A correspondingly large viomycin-dependent stabilization of near-cognate TC in the very same late binding state would readily explain the viomycin-induced increase in kinetic efficiency of the near-cognate reaction (*Figure 2*) and the decrease in dissociation rate of TC in the cognate reaction (*Figure 2A*) (*Carter et al., 2000*).

## A kinetic model for inhibition of translational fidelity by viomycin

As shown above, viomycin binding stabilizes both cognate and near-cognate TC on the ribosome but increases the kinetic efficiency only for near-cognate reactions, even though cognate TCs are frequently rejected by the ribosome. These observations can be accounted for by the existence of an initial binding state where any TC lacks codon·anticodon interaction (*Figure 3*) in accordance with previous work on initial selection (*Geggier et al., 2010*; *Gromadski and Rodnina, 2004*; *Loveland et al., 2017*; *Pape et al., 1998*; *Pavlov and Ehrenberg, 2018*; *Zhang et al., 2018*). In this case, cognate and near-cognate TCs have equal probability of dissociating from the ribosome rather than proceeding to formation of codon·anticodon contact in the decoding site and subsequent activation of the monitoring bases. In these latter states, near-cognate tRNA has high probability of moving backward to the preceding state while cognate tRNA has high probability of moving forward to the upcoming state. Viomycin binds to the state with activated monitoring bases, and when this happens any TC present in the A site is prevented from moving backwards, eventually leading to GTP hydrolysis by EF-Tu and subsequent peptide bond formation with 100% probability. The viomycin dependence of $k_{cat}/K_M$ for GTP hydrolysis in such a mechanism is given by (supplementary methods):

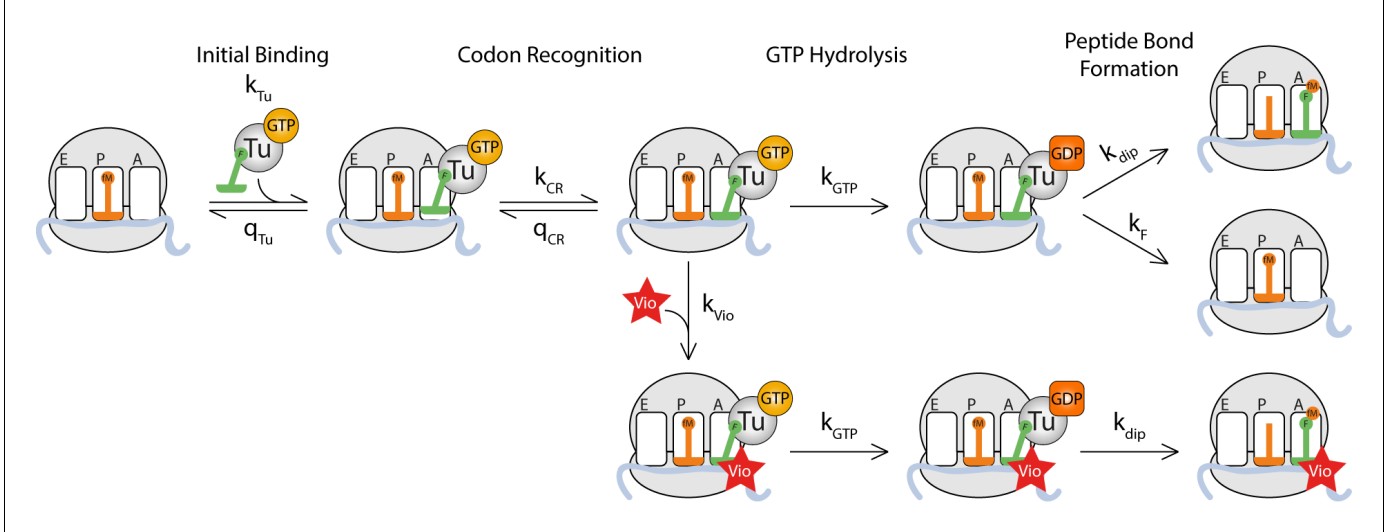

**Figure 3.** Kinetic model for viomycin action during tRNA selection. TC binds to a ribosome with an empty A site with rate constant $k_1$ to form a viomycin-insensitive initial binding complex where the codon·anticodon interaction is not yet established. From this state either the TC dissociates with rate constant $q_2$ or the ribosome proceeds along the selection pathway, with rate constant $k_2$ to codon anticodon contact. From this state the ribosome can either return to the initial binding state with rate constant $q_3$ or proceed to a viomycin-sensitive state, with rate constant $k_3$, where the codon·anticodon interaction is monitored by the activated monitoring bases A1492 and A1493. In this state three events can take place; the ribosome can return to the previous state with rate constant $q_4$, the ribosome can proceed with hydrolysis of EF-Tu bound GTP with rate constant $k_4$ or viomycin can bind with rate constant $k_{vio}$. After GTP hydrolysis viomycin-free ribosomes can either form a peptide bond with rate constant $k_5$ or reject the tRNA in the A site with rate constant $k_F$. Viomycin-bound ribosomes are unable to reject the tRNA present in the A site and therefore will always proceed with GTP hydrolysis and peptide bond formation regardless of the nature of the codon·anticodon interaction.
DOI: https://doi.org/10.7554/eLife.46124.004

$$\left( k_{cat}/K_M \right)^{c,nc} = \frac{k_1}{1 + \frac{q_2}{k_2}\left( 1 + \frac{q_3^{c,nc} q_4^{c,nc}}{k_3(k_4 + k_V[V])} \right)} \tag{1}$$

which for cognate substrates simplifies to (supplementary methods):

$$\left( \frac{k_{cat}}{K_M} \right)^c = \frac{k_1}{1 + \frac{q_2}{k_2}} \tag{2}$$

Here $k_1$ and $q_2$ are the rate constants for ternary complex association to and dissociation from the initial binding state $C_2$. Parameters $k_2$ and $q_3^{c,nc}$ are the rate constants for entry into and return from the first codon recognition state, $C_3$. Parameters $k_3$ and $q_4^{c,nc}$ are the rate constants for entry into and return from the second codon recognition state, $C_4$ where $k_4$ is the rate constant for GTP hydrolysis by EF-Tu (*Figure 3*). The suffixes *c* and *nc* denote parameters that vary between cognate and near-cognate reactions, respectively. With a cognate codon·anticodon interaction $q_3^c q_4^c/(k_3 k_4)$ is expected to be much smaller than one, due to small backward and large forward rate constants (*Pavlov and Ehrenberg, 2018*; *Zhang et al., 2018*), which is what gives rise to the simplified expression for the $k_{cat}/K_M$ (*Equation 2*). With a near-cognate codon·anticodon interaction $q_3^{nc} q_4^{nc}/(k_3 k_4)$ is expected to be much larger than one, due to comparatively large backward rate constants (*Pavlov and Ehrenberg, 2018*; *Zhang et al., 2018*). This leads to much larger $k_{cat}/K_M$ for cognate than for near-cognate reactions and thereby high accuracy. It also explains why the cognate $k_{cat}/K_M$ is insensitive to viomycin while the near-cognate $k_{cat}/K_M$ increases sharply with increasing drug concentration leading to sharply decreasing accuracy.

It follows from *Equation 1* that in the presence of viomycin the normalized accuracy, A, for initial codon selection, defined as the ratio between cognate and near-cognate $k_{cat}/K_M$- values for GTP hydrolysis, is approximated by:

$$A = \frac{1 + \frac{q_2}{k_2}\left(1 + \frac{q_3^{nc} q_4^{nc}}{k_3(k_4 + k_V[V])}\right)}{1 + \frac{q_2}{k_2}} \tag{3}$$

It is clear from *Equation 3* that the value of $A$ depends on the ratio between the selective back reaction product $q_3^{nc} q_4^{nc}$ and the non-selective forward rate constant $k_3$ multiplied by a non-selective total forward rate constant, $k_4 + k_V[V]$, leading directly or *via* viomycin binding to GTP hydrolysis. As explained further in the Discussion and supplementary materials, we have assumed rate constants $k_4$ and $k_V$ to be the same in cognate and near-cognate cases. This is supported by crystal (*Demeshkina et al., 2012*; *Demeshkina et al., 2013*) and cryo-EM (*Fislage et al., 2018*; *Loveland et al., 2017*) structures showing virtually identical geometries for cognate and near-cognate codon·anticodon interactions as well as kinetics data (*Zhang et al., 2018*). It follows directly from *Equation 3* that as long as $k_V[V] \gg k_4$ we can write (*Geggier et al., 2010*; *Gromadski and Rodnina, 2004*; *Johansson et al., 2008*; *Pape et al., 1998*; *Wohlgemuth et al., 2010*) (supplementary methods):

$$(k_{cat}/K_M)^{nc} = (k_{cat}/K_M)^c \cdot \frac{[V]}{[V] + K_I^{nc}}, \tag{4}$$

where $K_I^{nc}$ is the viomycin concentration at which the accuracy, $A$, has decreased to just two:

$$K_I^{nc} = \frac{q_2}{q_2 + k_2} \cdot \frac{q_3^{nc} q_4^{nc}}{k_3 k_V} \tag{5}$$

By fitting of *Equation 4* to the experimental data points in *Figure 1B* and *Figure 1D* the $K_I$ value for $\mathrm{tRNA}_{\mathrm{GAA}}^{\mathrm{Phe}}$ reading the near-cognate codon CUC was estimated as $(3.1 \pm 0.2)$ mM from GTP hydrolysis (*Figure 1B*) and as $(3.3 \pm 0.1)$ mM from dipeptide formation (*Figure 1D*). While these concentrations may appear high, it should be noted that at a viomycin concentration equal to $K_I$ the near cognate $k_{cat}/K_m$ parameter is half that of the cognate one meaning that accuracy has been reduced to only 2. The ribosome would be unable to produce functional proteins even at far lower drug concentrations. Another type of $K_I$ value can be estimated for the cognate reaction from the chase experiment data in *Figure 2B* by linear fitting of the following expression (supplementary methods):

$$\tau_{diss} = \tau_{diss}^0 + \frac{1}{q_V}\left(1 + \frac{[V]}{K_I^c}\right), \tag{6}$$

Here, the first term, $\tau_{diss}^0$, is a contribution to the mean time of TC dissociation that remains unaltered as the fraction of viomycin-bound ribosomes increases from zero to 100%. Parameter $q_V$ is the rate constant for viomycin dissociation from the ribosome and $K_I^c$ is the viomycin concentration at which the rate of viomycin rebinding to a ribosome with bound TC is equal to the rate of TC dissociation when unhindered by rebinding of viomycin. Note that all ribosomes are assumed to be viomycin bound in *Equation 6*. It follows that $K_I^c$ is given by (supplementary material)

$$K_I^c = \frac{q_2}{q_2 + k_2} \cdot \frac{q_3^c q_4^c}{k_3 k_V} \tag{7}$$

This gives a $K_I^c$ value for $\mathrm{tRNA}_{\mathrm{GAA}}^{\mathrm{Phe}}$ reading the cognate codon UUC of $(9.4 \pm 3.3)$ µM.

From these expressions, it can be seen that $K_I$ increases when the compounded back rate constant for rejection of tRNA from the codon recognition states, $q_3^{c/nc} q_4^{c/nc}$, increases. This is because larger values of $q_3^{c/nc} q_4^{c/nc}$ leave a smaller time window for viomycin to bind before the tRNA is rejected. $K_I$ decreases when the ratio $q_2/k_2$ decreases. Small values of this ratio mean that each time a TC returns to the non-selective initial binding state it has a larger probability to return to the codon-selective states for GTPase activation, affording viomycin multiple chances to bind. Note that $K_I$ is completely insensitive to the rate constant for GTP hydrolysis $k_4$.

## The viomycin sensitivity of a mismatched codon·anticodon pair correlates strongly with the accuracy of initial codon selection

The model presented above predicts that viomycin sensitivity ($K_I^{nc}$) for any codon·anticodon pair depends on the accuracy of initial codon selection as defined by the product of the near-cognate back reaction rate constants $q_3^{nc}$ and $q_4^{nc}$ (*Equations 4 and 5*) in the absence as well as presence of viomycin. For different codon·anticodon pairs initial codon selectivity varies over more than two orders of magnitude (*Johansson et al., 2012*; *Zhang et al., 2015*) implying that viomycin sensitivity could also vary significantly from pair to pair. To test this prediction, we measured how the $k_{cat}/K_M$ of dipeptide formation varied with viomycin concentration for $\text{tRNA}_{\text{GAA}}^{\text{Phe}}$ reading three additional near-cognate codons (*Figure 4A*); AUC, UAC or UUA (the underlined base differs from the cognate codon UUC). We quantified the viomycin sensitivity of each codon by estimating its $K_I$ value by fitting of *Equation 4* to plots of $k_{cat}/K_M$ versus viomycin concentration (*Figure 4B*). We also estimated the accuracies of initial selection by measuring $k_{cat}/K_M$ for GTP hydrolysis with the three codons in the absence of viomycin. Both sets of experiments were carried out exactly as described above. This gave $K_I$ values of 7 ± 0.3 mM for AUC, 29.0 ± 2.7 mM for UAC and 3.25 ± 0.20 mM for UUA and accuracies of initial selection of 2400 ± 120 for AUC, 6400 ± 670 for UAC and 580 ± 47 for UUA.

Plotting the accuracies of initial selection versus the $K_I$ values shows a clear correlation between the two (*Figure 4C*). This indicates that differences in viomycin sensitivity and accuracy of initial selection between different codon·anticodon pairs depend on differences in the same elemental rate constants. This is in line with the hypothesis that the rate constant $k_4$ is neutral to the cognate or near-cognate nature of the codon·anticodon interaction and that accuracy only varies with the product $q_3^{nc} q_4^{nc}$ (*Equation 3*) (*Geggier et al., 2010*; *Thompson, 1988*; *Gromadski and Rodnina, 2004*; *Pape, 1999*).

## A model to quantify viomycin-induced translational errors

We can now construct a model to estimate the frequency of extra missense errors induced by viomycin during translation. The probability that a given tRNA is trapped by viomycin on a ribosome

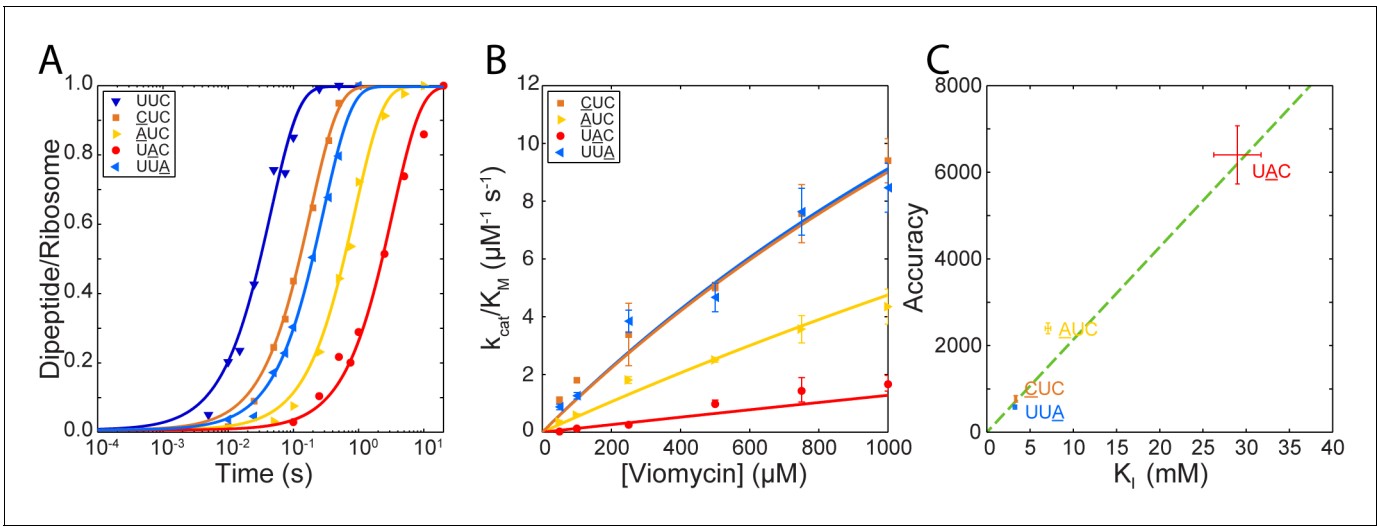

**Figure 4.** Correlation between initial selection accuracy and viomycin sensitivity for four codon· anticodon pairs. (A) Time courses of f[³H]Met-Phe dipeptide formation for 1 μM Phe-tRNA$^{\text{Phe}}_{\text{GAA}}$ containing TCs reacting with 70S ribosomes, displaying either a cognate UUC codon or one of the near-cognate codons CUC, AUC, UAC or UUAin the A site (the underlined base differs from the cognate codon), in the presence of 250 μM viomycin. Solid lines represent fits of single exponential equations to the data. (B) of dipeptide formation by Phe-tRNA$^{\text{Phe}}_{\text{GAA}}$ containing TCs reacting with 70S ribosomes, displaying the near-cognate codons CUC, AUC, UAC or UUAin the A site, at varying concentrations of viomycin. Solid lines represent fits of *Equation 3* to the data. (C) The accuracy of initial selection for Phe-tRNA$^{\text{Phe}}$containing TCs reading the indicated codons plotted against the concentration of viomycin required to increase the efficiency of each near-cognate reaction to half that of the cognate reaction ( ). The dotted line is a linear regression of the data illustrating the correlation predicted by *Equation 2*. All error bars represent SEM.
DOI: https://doi.org/10.7554/eLife.46124.005

displaying a given codon in the A site will depend on the intracellular concentration of that tRNA species as well as on how long it remains bound in the ribosomal A site during initial selection. This leads to the following expression for the probability that a missense error is caused by viomycin (supplementary methods):

$$P(E) = \frac{1}{1 + \frac{[T_3]^c}{\sum_k [T_3]_k^{nc} \frac{[Vio]}{[Vio] + K_{Ik}}}} \tag{8}$$

Here, $[T_3]_k^{nc}$ is the concentration of near- or non-cognate TC species $k$, $[T_3]^c$ is the cognate TC concentration and $K_{Ik}$ is the $K_I$ value for tRNA of type $k$ reading the codon in the A site. As an example, from our in vitro experiments; 80 ± 8 nM viomycin is required to double the rate at which tRNA^Phe reads the codon C̲UC; this would roughly double the error rate assuming that there were equal concentrations of UUC and C̲UC displaying ribosomes in the reaction mixture.

## Discussion

Based on the results presented above, we have constructed a kinetic model for how viomycin reduces the fidelity of mRNA decoding (*Figure 3*). We also show that the loss of translational fidelity due to viomycin is to a good approximation governed by a single kinetic parameter ($K_I^{nc}$) for each codon·anticodon pair, and we have precisely determined its value for four such pairs. In this model, when a ternary complex first binds to the ribosome the codon·anticodon interaction is not yet established, the ribosome is not yet sensitive to viomycin and the monitoring bases are inactive. Subsequent establishment of codon·anticodon interaction and activation of the monitoring bases then leads to a highly selective ribosomal state to which viomycin can bind. The viomycin sensitivity of a ribosome with a given codon·anticodon pair in the A site is defined by the $K_I^{nc}$-value, which depends on how much time the ribosome spends in this viomycin-sensitive 'codon recognition' state before TC dissociates (*Equation 5*). Viomycin binding to this state effectively traps the tRNA present in the A site, abolishing the ability of the ribosome to discriminate between cognate and non-cognate tRNAs in both initial selection and proofreading selection, committing the viomycin-bound ribosome to GTP hydrolysis and peptide bond formation with 100% probability.

It has been suggested that most of the high accuracy of translation is achieved through larger forward rate constants of GTP hydrolysis and tRNA accommodation for cognate than for near- and non-cognate substrates (*Gromadski and Rodnina, 2004*; *Pape, 1999*). Such a mechanism would imply that a large part of the variation in the accuracy of initial selection between different mismatched codon·anticodon pairs comes from variation of the rate of GTP hydrolysis rather than from variation of the tRNA rejection rate. More recently, high -resolution ribosome structures from crystallography (*Demeshkina et al., 2012*; *Demeshkina et al., 2013*) and cryo-EM (*Fislage et al., 2018*; *Loveland et al., 2017*) have revealed that cognate and near-cognate codon-anticodon complexes from tRNAs (*Demeshkina et al., 2012*; *Demeshkina et al., 2013*) or TCs (*Loveland et al., 2017*) have very similar structures. This suggests the existence of a highly selective state in which cognate and near-cognate TCs have the same orientation in the A/T state and, by inference, the same rate constant for GTP hydrolysis. This suggestion of a codon·anticodon insensitive rate constant $k_4$ is fully compatible with earlier kinetics results showing that the maximal rate of GTP hydrolysis ($k_{cat}$) is lower in near-cognate than cognate cases and becomes equal upon addition of aminoglycosides (*Pape, 1999*; *Pavlov and Ehrenberg, 2018*; *Zhang et al., 2018*). We suggest that in those earlier studies the increase in the Michaelis-Menten parameter $k_{cat}$, due to decreasing back reaction rate constants on drug addition, was instead mistakenly interpreted (*Gromadski and Rodnina, 2004*; *Pape, 1999*) as an increase of the catalytic rate constant, $k_4$, for GTP hydrolysis (*Pavlov and Ehrenberg, 2018*; *Zhang et al., 2018*).

Furthermore, our observation of a strong correlation between the accuracy of initial codon selection in the absence of viomycin and the viomycin sensitivity ($K_I^{nc}$) (*Figure 4C*) is fully in line with the present hypothesis of a codon-anticodon insensitive rate constant for GTP hydrolysis. This type of correlation requires that virtually all the variation in accuracy between different codon·anticodon pairs comes from variation of tRNA rejection rates rather than from variation of GTP hydrolysis rates. If all non-cognate tRNAs remained on the ribosome for approximately the same amount of time and

accuracy was primarily determined by their propensity to trigger GTP hydrolysis, that is by variation in $k_4$ (*Gromadski and Rodnina, 2004*), we would observe approximately the same $K_I^{nc}$ for all codon·anticodon pairs (*Equation 5*), which is not the case (*Figure 4C*).

In all available structures of viomycin-bound ribosomes the drug is positioned between rRNA helices h44 and H69 in the space vacated by the bases A1492 and A1493 when they flip out to interact with the codon·anticodon minihelix (*Brilot et al., 2013*; *Pulk and Cate, 2013*; *Stanley et al., 2010*; *Zhou et al., 2012*). Of these structures two contain an A-site tRNA (*Brilot et al., 2013*; *Stanley et al., 2010*) and in both cases it is a cognate tRNA; leaving open the question of how viomycin can bind rapidly to a ribosome with a non-cognate tRNA where A1492 and A1493 are thought to occupy the drug binding site (*Carter et al., 2000*). Rapid binding of viomycin to non-cognate ribosome·tRNA complexes is explained by recent observations that A1492 and A1493 flip out after initial binding of a tRNA to the A site regardless of Watson-Crick base pairing between the codon and the anticodon (*Loveland et al., 2017*). The less a given codon·anticodon helix can be stabilized by interactions with A1492 and A1493 the less energetically favorable the flipped-out conformation becomes. Thus, the more easily a tRNA can be rejected by the ribosome the less time A1492 and A1493 spend in their flipped-out conformation. This link between the accuracy and the length of the time window during which the viomycin binding site is open explains our observed correlation between accuracy and viomycin sensitivity. Thus, it is likely that the bases A1492 and A1493 rapidly fluctuate between their active flipped-out and inactive flipped-in conformations when any tRNA is present in the A site. Such a model has been suggested previously from studies of A-site dynamics in the absence of tRNA (*Fourmy et al., 1998*; *Sanbonmatsu, 2006*; *Vaiana and Sanbonmatsu, 2009*) and recently based on structural studies of tRNA selection by both mammalian and bacterial ribosomes (*Fislage et al., 2018*; *Loveland et al., 2017*; *Shao et al., 2016*). In particular, *Loveland et al. (2017)* shows that flipping-out of A1492 and A1493 happens early in decoding for both cognate and near-cognate tRNAs, binding of viomycin would then force A1492 and A1493 to remain in their flipped-out positions, leading to activation of G530, followed by closure of the 30S subunit and GTP hydrolysis. The complete absence of proofreading selection by viomycin-bound ribosomes further implies that the A1492 and A1493 play a role also in this process and that proofreading may be mediated by the same conformational changes of the decoding center as initial selection.

*Equation 6* provides a model to evaluate the probability for a given viomycin-induced missense error at any codon. To fully evaluate this expression for the situation in a living cell, it is necessary to know the concentration of all tRNA species as well as the $K_I^{nc}$ values for all codon·anticodon pairs. However, some conclusions can be drawn even without this information. The viomycin-induced error frequency is large when the concentration of cognate tRNA is small and when there is a high concentration of near- or non-cognate tRNAs that are not efficiently discriminated against during initial selection. These are the same conditions that cause naturally occurring translational error hot-spots, implying that in the cell viomycin primarily acts to enhance such pre-existing hot-spots. Further, since proofreading selection is completely disabled on viomycin-bound ribosomes, it is unable to carry out its suggested function in neutralizing error hot-spots in initial selection (*Zhang et al., 2016*). This means that viomycin will alter not just the overall frequency of translational errors but also their distribution.

The $K_I^{nc}$ values estimated here are remarkably *large* considering how little viomycin is required to significantly reduce the rate of translocation (*Holm et al., 2016*) but direct comparison of the error-induction and translocation inhibition effects of viomycin is difficult as it is largely unknown how changes in the translational error rate affect bacterial growth rate. From the available data (*Hughes, 1991*), it seems that small changes in translational fidelity cause significantly smaller changes in growth rate than what is caused by comparable changes in translation speed. Given the parameter estimates in this and our previous study on translocation inhibition by viomycin (*Holm et al., 2016*), the error-inducing effect of the drug is likely responsible for only a small fraction of its antimicrobial activity under typical laboratory conditions, but may be more important in the clinical setting. The clinical target of the tuberactinomycins, the slow growing *M. tuberculosis*, normally maintains a smaller number of ribosomes per cell compared to faster growing bacteria (*Cox, 2003*). It could therefore potentially significantly reduce the effectiveness of translation speed inhibition, but not inhibition of translational accuracy, by overproduction of ribosomes (*Dennis, 1976*; *Feldman et al., 2010*; *Maitra and Dill, 2016*).

Antibiotics of the aminoglycoside class bind to the ribosome in a site that partially overlaps with that of viomycin (*Carter et al., 2000*; *Stanley et al., 2010*), suggesting that aminoglycosides and viomycin have overlapping modes of action. The detailed effects of three types of aminoglycosides on the accuracy of tRNA selection were recently investigated in a study (*Zhang et al., 2018*) which, together with the present study, clarifies differences and similarities of the modes of action of these two groups of antibiotics. Unlike viomycin, aminoglycosides bind to the ribosome with high affinity independently of the presence of an A-site-bound tRNA or ternary complex (*Feldman et al., 2010*; *Pape et al., 2000*; *Zhang et al., 2018*). Like viomycin, aminoglycosides alter the equilibrium between the active and inactive conformations of the monitoring bases A1492 and A1493, although the aminoglycoside-induced equilibrium shift is much smaller (*Fislage et al., 2018*; *Zhang et al., 2018*) than that of viomycin. Thus, the modes of action of these two classes of drugs are structurally similar in that they both force the ribosome into a state where the A-site tRNA is stabilized by activation of the monitoring bases. At the same time, their modes of action are kinetically distinct, since the two drugs bind to the ribosome during different stages of the ternary complex selection process and viomycin must be present at a much higher concentration for effective error induction than an aminoglycoside.

In summary, we have provided a quantitative kinetic model for the error-inducing effect of viomycin which together with our previous study on translocation inhibition (*Holm et al., 2016*) covers both known functions of the tuberactinomycin antibiotics. The model for initial selection of tRNA by the ribosome and the function of the monitoring bases suggested by our results is strongly supported by recent structural studies (*Loveland et al., 2017*; *Shao et al., 2016*) and calls into question prevailing ideas of how the high accuracy of translation is achieved. Our models and methods can be used to characterize the antimicrobial mechanisms of other tuberactinomycins and potential new tuberactinomycin derivatives and to understand the mechanisms of tuberactinomycin resistance mutations, which is highly relevant in terms of treatment of tuberculosis and related diseases.

# Materials and methods

## Buffers and reagents

All experiments were performed at 37°C in HEPES-polymix buffer (95 mM KCl, 5 mM $NH_4Cl$, 0.5 mM $CaCl_2$, 8 mM putrescine, 1 mM spermidine, 5 mM Mg(OAc)$_2$, 1 mM dithioerythritol and 5 mM HEPES pH 7.5). All reaction mixes contained 10 mM phosphoenolpyruvate (PEP), 1 µg/ml pyruvate kinase (PK) and 0.1 µg/ml myokinase (MK). His-tagged initiation factors IF1, IF2 and IF3, elongation factor Ts, and phenylalanine and leucine aminoacyl tRNA-synthetases were purified using nickel-affinity chromatography (HisTrap GE Healthcare). Wild-type elongation factor Tu was prepared as in *Ehrenberg et al. (1990)*. All protein concentrations were determined using the Bradford assay. Ribosomes (*E. coli* MRE600) and f[$^3$H]Met-tRNA$^{fMet}$ were prepared according to *Antoun et al. (2004)*; ribosome concentration was determined spectrophotometrically. XR7 mRNAs with coding sequences AUG-UUC, AUG-CUC, AUG-UAC and AUG-UUA were prepared as in *Borg and Ehrenberg (2015)*, see theGV Appendix I for full mRNA sequences. tRNA$^{Phe}$ was prepared as in *Holm et al. (2016)*. [$^3$H]Met and [$^3$H]GTP were from Perkin-Elmer, viomycin was from USP, all other chemicals were from either Merck or Sigma-Aldrich.

## Construction and purification of EF-Tu$^{H84A}$

The wild type *tufA* gene from *E. coli* Mg1655 was cloned in the pET21b vector with a C-terminal hexahistidine tag. Using this construct, the Histidine at position 84 was changed to Alanine by following the standard protocol from the QIAGEN site directed mutagenesis kit. Successful mutation was confirmed by DNA sequencing. His-tagged EF-Tu$^{H84A}$ was overexpressed in *E. coli* BL21(DE3) and purified using nickel-affinity chromatography (HisTrap GE Healthcare). The identity and purity of the H84A mutant protein was confirmed by mass spectrometry.

## GTP-hydrolysis experiments

Two mixtures were prepared. The ribosome mixture contained 70S ribosomes (1.0–2.0 µM), IF1, IF2 and IF3 (2 µM each), fMet-tRNA$^{fMet}$(1.5–3.0 µM), mRNA (3 µM), GTP (1 mM) and ATP (1 mM). The TC mixture contained EF-Tu (0.3–0.6 µM), phenylalanine (200 µM), PheRS (0.5 µM), tRNA$^{Phe}$ (2 µM),

viomycin (0–2000 µM), [³H]GTP (0.3–0.6 µM) and ATP (2 mM). After 15 min incubation at 37˚C, equal volumes of the two mixes were rapidly mixed and the reaction quenched at different time points with formic acid (17% final concentration) using a quench-flow instrument (RQF-3 KinTek corp.). After quenching, the samples were centrifuged at 20,800 g. The supernatant, containing the [³H] GTP and [³H]GDP was analyzed by anion exchange chromatography with on-line scintillation counting (β-RAM model 4 IN/US systems). A Mono-Q GL column (GE Healthcare) was used and the mobile phase was a multistep gradient of 0–2 M NaCl in 20 mM Tris (pH 7.5).

### Dipeptide formation experiments

Two mixtures were prepared. The ribosome mixture contained 70S ribosomes (0.5 µM), IF1, IF2 and IF3 (1 µM each), f[³H]Met-tRNA$^{fMet}$ (1 µM), mRNA (2 µM), GTP (1 mM) and ATP (1 mM). The TC mixture contained EF-Tu (1–10 µM), EF-Ts (1 µM), phenylalanine (200 µM), PheRS (0.5 µM), tRNA$^{Phe}$ (12 µM), viomycin (0–2000 µM), GTP (1 mM) and ATP (1 mM). After 15 min incubation at 37˚C, equal volumes of the two mixes were rapidly mixed and the reaction quenched at different time points with formic acid (17% final concentration) using a quench-flow instrument (RQF-3 KinTek corp.). After quenching, the samples were centrifuged at 20,800 g and the supernatant discarded. The pellet was dissolved in 165 µl 0.5 M KOH to cleave the peptides from the tRNA. After 10 min 13 µl of 100% formic acid was added, the samples were centrifuged at 20,800 g and the radioactive peptides in the supernatant were analyzed by RP-HPLC using a H₂O/MeOH/trifluoroacetic acid (58/42/0.1 by volume) mobile phase and a C-18 column (Merck) with on-line scintillation counting (β-RAM model 4 IN/US systems) to quantify the relative amounts of f[³H]Met and f[³H]Met-Phe.

### EF-Tu$^{H84A}$ chase experiments

Three mixtures were prepared. The ribosome mixture contained 70S ribosomes (0.75 µM), IF1, IF2 and IF3 (1 µM each), f[³H]Met-tRNA$^{fMet}$ (1 µM), mRNA (2 µM), GTP (1 mM) and ATP (1 mM). The first TC mixture contained EF-Tu$^{H84A}$ (15 µM), phenylalanine (200 µM), PheRS (0.5 µM), tRNA$^{Phe}$ (15 µM), viomycin (0–400 µM), GTP (1 mM) and ATP (1 mM). The second TC mixture contained EF-Tu (1.5 or 12 µM), EF-Ts (1 µM), phenylalanine (200 µM) or leucine (200 µM), PheRS (0.5 µM) or LeuRS (0.5 µM), tRNA$^{Phe}$ (2 µM) or bulk tRNA of which 2 µM was tRNA$^{Leu2}$ and an additional 10 µM were other leucine tRNA isoacceptors, viomycin (0–600 µM), GTP (1 mM) and ATP (1 mM). All three mixes were incubated at 37˚C for 15 min. During the experiment, one volume of the ribosome mixture was mixed with one volume of the first TC mixture, the resulting mixture was incubated for 5–10 s and then one volume of the second TC mixture was added. The reaction was quenched at different time points after the addition of the second TC mixture using formic acid (17% final). The samples were treated as the quench flow peptide samples above.

### Data analysis and curve fitting

Reaction rates were estimated by fitting of single exponential functions to the experimental time courses except for the GTP hydrolysis reactions without viomycin on near-cognate codons which were analyzed as in *Johansson et al. (2012)*. $k_{cat}/K_M$ values were estimated by fitting of the Michaelis-Menten equation to plots of reaction rates versus concentration. The linear regression in *Figure 4C* was based on the method in *York et al. (2004)*. All curve-fittings were implemented in Matlab R2015b. For derivations of the equations used in the main text see Appendix I.

## Acknowledgements

This work is funded by the research grants from the Swedish Research Council (2018–05498 (NT), 2016–06264 (Research Environment)), Carl-Tryggers Foundation (CTS 18: 338), Wenner-Gren Foundation (UPD2017-0238) and the Knut and Alice Wallenberg Foundation (KAW 2011.0081 to Ribo-CORE and KAW 2017.0055) to Suparna Sanyal. The authors also thank Raymond Fowler for expert technical assistance for the purification of translation factors.

## Additional information

### Funding

| Funder | Grant reference number | Author |
|---|---|---|
| Swedish Research Council | 2018-05498 (NT) | Suparna Sanyal |
| Carl Tryggers Stiftelse för Vetenskaplig Forskning | CTS 18: 338 | Suparna Sanyal |
| Wenner-Gren Foundation | UPD2017-0238 | Suparna Sanyal |
| Knut och Alice Wallenbergs Stiftelse | KAW 2011.0081 to RiboCORE | Suparna Sanyal |
| Knut och Alice Wallenbergs Stiftelse | KAW 2017.0055 | Suparna Sanyal |
| Swedish Research Council | 2016-06264 (Research Environment) | Suparna Sanyal |

The funders had no role in study design, data collection and interpretation, or the decision to submit the work for publication.

### Author contributions

Mikael Holm, Conceptualization, Data curation, Formal analysis, Validation, Methodology, Writing—original draft, Writing—review and editing; Chandra Sekhar Mandava, Resources, Investigation, Writing—review and editing; Måns Ehrenberg, Formal analysis, Writing—review and editing; Suparna Sanyal, Conceptualization, Resources, Formal analysis, Supervision, Funding acquisition, Investigation, Project administration, Writing—review and editing

### Author ORCIDs

Mikael Holm https://orcid.org/0000-0002-4361-9554
Chandra Sekhar Mandava https://orcid.org/0000-0002-3028-3270
Suparna Sanyal https://orcid.org/0000-0002-7124-792X

### Decision letter and Author response

Decision letter https://doi.org/10.7554/eLife.46124.009
Author response https://doi.org/10.7554/eLife.46124.010

## Additional files

### Supplementary files

• Transparent reporting form
DOI: https://doi.org/10.7554/eLife.46124.006

### Data availability

All data generated or analysed during this study are included in the manuscript.

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

## Appendix 1

DOI: https://doi.org/10.7554/eLife.46124.007

### mRNA sequences

Complete nucleotide sequences for the mRNA molecules used. The start codon and A-site codon are in bold.

UUC (cognate for tRNA[Phe])
5'-GAAUUCGGGCCCUUGUUAACAAUUAAGGAGGUAUUAA**AUGUUC**UCUAA UUGCAGAAAAAAAAAAAAAAAAAAAAAA-3'

AUC (near-cognate, first position mismatch)
5'-GAAUUCGGGCCCUUGUUAACAAUUAAGGAGGUAUUAA**AUGAUC**UCUAA UUGCAGAAAAAAAAAAAAAAAAAAAAAA-3'

CUC (near-cognate, first position mismatch)
5'-GAAUUCGGGCCCUUGUUAACAAUUAAGGAGGUAUUAA**AUGCUC**UCUAA UUGCAGAAAAAAAAAAAAAAAAAAAAAA-3'

UAC (near-cognate, second position mismatch)
5'-GAAUUCGGGCCCUUGUUAACAAUUAAGGAGGUAUUAA**AUGUAC**UCUAA UUGCAGAAAAAAAAAAAAAAAAAAAAAA-3'

UUA (near-cognate, third position mismatch)
5'-GAAUUCGGGCCCUUGUUAACAAUUAAGGAGGUAUUAA**AUGUUA**UCUAA UUGCAGAAAAAAAAAAAAAAAAAAAAAA-3'

### Data analysis and curve fitting

#### Introduction to the four-step scheme for initial codon selection

A four-step scheme for viomycin action on initial codon selection by transfer RNA in ternary complex with EF-Tu and GTP may be formulated as (*Loveland et al., 2017*; *Pavlov and Ehrenberg, 2018*; *Zhang et al., 2018*):

$$R_1 + T_3 + Vio \underset{q_2}{\overset{k_1}{\rightleftharpoons}} C_2 + Vio \underset{q_3^x}{\overset{k_2}{\rightleftharpoons}} C_3 + Vio \underset{q_4^x}{\overset{k_3}{\rightleftharpoons}} C_4 + \overset{k_4}{\rightarrow}$$

$$(q_V) \uparrow\downarrow k_V[V]$$

$$C_4 + \overset{k_{4V}}{\rightarrow}$$

(1)

$R_1$ is the ribosome with initiator tRNA in the P site and an empty A site programmed with a codon cognate (x=c) or near-cognate (x=nc) to the Phe-tRNA[Phe] containing ternary complex, $T_3$. In state $C_2$, codon•anticodon contact has yet to be established. In $C_3$, there is codon-anticodon contact (*Zhang et al., 2016*) with partially inactive monitoring bases (*Loveland et al., 2017*) and an open 30S subunit. In $C_4$, all monitoring bases (A1492, A1493 and G530) are active and the 30S subunit is closed (*Fislage et al., 2018*; *Loveland et al., 2017*; *Zhang et al., 2018*). Complex $C_3$ moves to state $C_4$ with rate constant k3. State $C_4$ is subjected to GTP hydrolysis with rate constant, moves backward to state $C_3$ with rate constant or moves to state $C_{4V}$ by binding to viomycin with compounded rate constant. State $C_{4V}$ is either subjected to GTP hydrolysis with rate constant, butmay return to state $C_4$ through very slow viomycin dissociation. Therefore, viomycin dissociation from $C_{4V}$ is significant only in chase experiments with a GTPase-deficient mutant of EF-Tu, as described below and in *Gromadski and Rodnina (2004)* and *Zhang et al. (2018)*. In scheme 1, binding of viomycin occurs only to complex $C_4$, in line with the proposal that the monitoring bases are inactive (*Zhang et al., 2016*) or partially active in state $C_3$ (*Fislage et al., 2018*; *Loveland et al., 2017*) (see also Discussion). To account for all present data, we also make the assumption that structures $C_3$ and $C_4$ are comparatively rapidly equilibrating (*Pavlov and Ehrenberg, 2018*) (see also Discussion), so that:

$$R_1 + T_3 + Vio \underset{q_2}{\overset{k_1}{\rightleftharpoons}} C_2 + Vio \underset{q_{34}^x}{\overset{k_2}{\rightleftharpoons}} C_{34} + Vio \overset{k_{34}^x}{\rightarrow}$$
$$(q_V) \uparrow\downarrow k_V^x[V]$$
$$C_4 + \overset{k_{4V}}{\rightarrow}$$

(2)

Compounded parameters are related to the primary parameters in **Equation 1** through

$$q_{34}^x = q_3^x \cdot \frac{K_{34}^x}{K_{34}^x + 1}, k_{34}^x = k_4 \cdot \frac{1}{K_{34}^x + 1}, k_V^x = k_V \cdot \frac{1}{K_{34}^x + 1}, K_{34}^x = \frac{q_4^x}{k_3}$$

(3)

To derive Michaelis-Menten parameters for scheme 2, we use the mean-time approach (**Bilgin et al., 1992**; **Borg and Ehrenberg, 2015**; **Borg et al., 2016**), as described below.

## Derivation of k$_{cat}$/K$_m$-parameters in cognate and near-cognate cases

To derive the kcat/Km parameters for scheme 2, we use ordinary differential equations for the time evolution of probabilitiesand for ternary complex, T3, to be in ribosome bound states C2 and C34, respectively. The reaction starts in complex C2 (p2(0)=1) and continues until it ends by T3 dissociation from C2, GTP hydrolysis in viomycin binding to followed by GTP hydrolysis. We note that in this approximation viomycin binding to in the following step leads to GTP hydrolysis with 100% probability:

$$\begin{aligned}
\frac{dp_2}{dt} &= -(k_2 + q_2)p_2 + q_{34}^x p_{34} \\
\frac{dp_{34}}{dt} &= k_2 p_2 - (k_{34}^x + k_V^x[V] + q_{34}^x)p_{34} \\
p_2(0) &= 1, p_{34}(0) = 0.
\end{aligned}$$

(4)

Mean times for T3 being in state "i" are defined as (**Bilgin et al., 1992**; **Borg and Ehrenberg, 2015**; **Borg et al., 2016**):

$$\tau_i = \int_0^\infty pi(t)dt; \, \text{i} = 2 \, \text{or} \, 3/4$$

(5)

Integrating equation system **Equation 4** from zero to infinite time, one obtains algebraic equations for the mean times:

$$\begin{aligned}
1 &= (k_2 + q_2)\tau_2^x - q_{34}^x \tau_{34}^x \\
0 &= k_2 \tau_2^x - (k_{34}^x + q_{34}^x + k_V^x[V])\tau_{34}^x
\end{aligned}$$

(6)

The solution is

$$\begin{aligned}
\tau_2^x &= \frac{k_{34}^x + k_V^x[V] + q_{34}^x}{(q_2 + k_2)(k_{34}^x + k_V^x[V]) + q_2 q_{34}^x}, \\
\tau_{34}^x &= \frac{k_2}{(q_2 + k_2)(k_{34}^x + k_V^x[V]) + q_2 q_{34}^x}
\end{aligned}$$

(7)

In general, $k_{cat}/K_m$ is defined by the association rate constant, $k_1$, multiplied by the probability, $p_{GTP}^x$, that ternary complex is subjected to GTP hydrolysis rather than dissociation from the ribosome (**Johansson et al., 2008**):

$$\left(\frac{k_{cat}}{K_m}\right)^x = k_1 p_{GTP}^x = k_1 \tau_{34}^x (k_{34}^x + [V]k_V^x) = \frac{k_1}{1 + a_2(1 + a_{34}^x)}$$

(8)

where

$$\begin{aligned}
a_2 &= q_2/k_2, \\
a_{34}^x &= \frac{q_{34}^x K_{34}^x}{k_{34}^x + k_V^x[V]} = \frac{q_3^x q_4^x}{k_3(k_4 + k_V[V])}
\end{aligned}$$

(9)

In analogy with initial codon selection in the presence of aminoglycosides we define a 'current' accuracy, an 'effective' initial codon selection and an 'intrinsic' initial codon selection (*Pavlov and Ehrenberg, 2018*; *Zhang et al., 2018*). The '*current*' *accuracy*, A([V]), for initial codon selection as function of the viomycin concentration, we define as:

$$A([V]) = \frac{1 + a_2\left(1 + a_{34}^{nc}\right)}{1 + a_2\left(1 + a_{34}^{c}\right)} \approx \frac{a_2}{1 + a_2} a_{34}^{nc} = \frac{a_2}{1 + a_2} \frac{q_3^x q_4^x}{k_3(k_4 + k_V[V])} \tag{10}$$

The '*effective*' *initial codon selection* as function of the viomycin concentration, $d_{ev}$, we define as:

$$d_{\mathrm{ev}} = \lim A([V])_{(a_2 \to \infty)} = \frac{1 + a_{34}^{nc}}{1 + a_{34}^{c}} \approx a_{34}^{nc} = \frac{1}{k_3} \frac{q_3^{nc} q_4^{nc}}{k_3 k_4 + k_V[V]} \tag{11}$$

The '*intrinsic*' *initial codon selection* as function of the viomycin concentration, $D([V])$, we define as

$$D = \lim A([V])_{(a_{34}^{c} \to \infty)} = \frac{a_{34}^{nc}}{a_{34}^{c}} = \frac{q_3^{nc} q_4^{nc}}{q_3^{c} q_4^{c}} \tag{12}$$

Since $a_{34}^{nc}$ is inversely proportional to the viomycin concentration at high values it follows from *Equation 11* that viomycin decreases the effective initial codon selection and from *Equation 12* that the intrinsic initial codon selection remains unaltered at changing drug concentration. A proposition here is that viomycin and aminoglycosides act according to partially similar principles since also aminoglycoside addition reduces $d_e$ and leaves $D$ unaltered (*Zhang et al., 2018*). The fundamental assumption leading to *Equations 11 and 12* is that rate constant $k_4$ in *Equation 9* is the same in cognate and near-cognate cases and not much larger in the former cases as previously claimed (*Gromadski and Rodnina, 2004*; *Pape, 1999*). This point is further discussed in the next section below where the maximal rate of GTP hydrolysis ($k_{cat}$) has been defined by its proper average value (*Pavlov and Ehrenberg, 2018*). We note that when $k_V[V] \gg k_4$ the approximation

$$\left(\frac{k_{cat}}{K_m}\right)^{nc} = \left(\frac{k_{cat}}{K_m}\right)^{c} \frac{[V]}{[V] + K_I}, \tag{13}$$

where

$$K_I = \frac{a_2}{(1 + a_2)} \frac{q_3^{nc} q_4^{nc}}{k_V k_3} \tag{14}$$

is valid. Under this condition of high viomycin concentration, the ratio ($K_I^{nc1}/K_I^{nc2}$) between the $K_I$-values for two near-cognate reactions approximates the ratio ($A^{nc1}/A^{nc2}$) between the $A_I$-values for the corresponding current accuracies when estimated in the absence of viomycin (*Equation 10*):

$$\frac{K_I^{nc1}}{K_I^{nc2}} = \frac{A^{nc1}}{A^{nc2}} = \frac{q_3^{nc1} q_4^{nc1}}{q_3^{nc2} q_4^{nc2}} \tag{15}$$

as seen experimentally. The point here is that if the forward rate constant for GTP hydrolysis, $k_4$, were different in the two cases, then the similarity between the $K_I$- and A-ratios in *Equation 15* would be invalid.

## Derivation of $k_{cat}$-parameters in cognate and near-cognate cases

In this section, we use mean time analysis to derive expressions for the $k_{cat}$-values of scheme (2). The relevant differential equations for the scheme in *Equation 2* are in this case given by

$$\begin{aligned}
\frac{dp_2}{dt} &= -k_2 p_2 + q_{34}^x p_{34} \\
\frac{dp_{34}}{dt} &= k_2 p_2 - \left(k_{34}^x + k_V^x[V] + q_{34}^x\right) p_{34} \\
\frac{dp_4}{dt} &= k_V^x[V] p_{34} - k_{4V} p_4 \\
p_2(0) &= 1, p_{34}(0) = 0, p_4(0) = 0.
\end{aligned} \tag{16}$$

Integration from zero to infinite time leads to the following algebraic equations:

$$\begin{aligned}
1 &= k_2 \tau_2 - q_{34}^x \tau_{34} \\
0 &= k_2 \tau_2 - \left(q_{34}^x + k_{34}^x + k_V^x[V]\right)\tau_{34} \\
0 &= k_V^x[V]\tau_{34} - k_{4V}\tau_4
\end{aligned} \tag{17}$$

The solution is

$$\begin{aligned}
\tau_2 &= \frac{1}{k_2}\left(1 + \frac{q_3^x q_4^x / k_3}{k_4 + k_V[V]}\right), \\
\tau_{34} &= \frac{1 + q_4^x / k_3}{k_4 + k_V[V]}, \\
\tau_{4V} &= \frac{1}{k_{4V}}\frac{k_V[V]}{k_4 + k_V[V]}.
\end{aligned} \tag{18}$$

The minimal time for GTP hydrolysis in ternary complex at saturating ribosome concentration, $\tau$, equal to the inverse of $k_{cat}$, corresponds to the sum of the times in *Equation 18*:

$$\tau = 1/k_{cat} = \tau_2 + \tau_{34} + \tau_{4V} \tag{19}$$

In cognate cases, the minimal reaction time is approximated by:

$$\tau = \frac{1}{k_{cat}} = \tau_2 + \tau_{34} + \tau_4 = \{k_{4V} = k_4\} = \frac{1}{k_2} + \frac{1}{k_4} \tag{20a}$$

As expected it is seen that the cognate $k_{cat}$-parameter does not respond to viomycin addition.

In near cognate cases, the minimal reaction time is given by:

$$\tau = \frac{1}{k_{cat}} = \{k_4 = k_{4V}\} = \frac{1}{k_2} + \frac{1}{k_4} + \frac{1}{(k_4 + k_V[V])}\left(\frac{q_3^{nc} q_4^{nc}}{k_2 k_3} + \frac{q_4^{nc}}{k_3}\right) \tag{20b}$$

Since $\frac{q_3^{nc} q_4^{nc}}{k_2 k_3} \gg 1$ it follows that $k_{cat}$ is much smaller in near-cognate than cognate cases and increases in proportion to increasing $k_V[V]$ when $k_V[V] \gg 1 \gg$ in spite of uniform and constant $k_4$-values. From this, we contend that the allegedly small near-cognate $k_4$-values estimated in the absence of drugs and their increase with aminoglycoside (*Gromadski and Rodnina, 2004*; *Pape, 1999*; *Pape, 1999*) or, as here, viomycin addition, reflect drug-induced variations in $k_{cat}$-values rather than $k_4$-values (*Pavlov and Ehrenberg, 2018*; *Zhang et al., 2018*).

## Effect of viomycin on the rate of dissociation of GTPase deficient ternary complex

Here, we adapt scheme 2 above to the chase experiments described in the main text, where a pre-bound, GTPase deficient ternary complex, $T_{3m}$, containing the EF-Tu mutant (*m*) H84A is chased by a native ternary complex, $T_3$:

$$R_1 + T_{3m} + Vio \leftarrow [q_2]C_{2m} + Vio \underset{q_{34}^x}{\overset{k_2}{\rightleftharpoons}} C_{34m} + Vio \tag{21}$$
$$(q_V) \uparrow\downarrow \mathrm{k}_V^x[V]$$
$$C_{4Vm}$$

$T_{3m}$ is pre-bound to the ribosome in either one of the states $C_{2m}$, $C_{34m}$ or $C_{4Vm}$ with probabilities $P_2^x(0)$, $P_{34}^x(0)$ or $P_{4V}^x(0)$, respectively, as determined by the stability of each $T_3$-bound complex:

$$
\begin{aligned}
P_2^x(0) &= 1 \Big/ \left(1 + \frac{k_2}{q_3^x} \cdot \frac{1+K_{34}^x}{K_{34}^x} + \frac{k_2}{q_3^x} \cdot \frac{k_V[V]}{q_V K_{34}^x}\right) = 1/N \\
P_{34}^x(0) &= \frac{k_2}{q_3^x} \cdot \frac{1+K_{34}^x}{K_{34}^x} \Big/ \left(1 + \frac{k_2}{q_3^x} \cdot \frac{1+K_{34}^x}{K_{34}^x} + \frac{k_2}{q_3^x} \cdot \frac{k_V[V]}{q_V K_{34}^x}\right) = \frac{k_2}{q_3^x} \cdot \frac{1+K_{34}^x}{K_{34}^x} / N \\
P_{4V}^x(0) &= \frac{k_2}{q_3^x} \cdot \frac{k_V[V]}{q_V K_{34}^x} \Big/ \left(1 + \frac{k_2}{q_3^x} \cdot \frac{1+K_{34}^x}{K_{34}^x} + \frac{k_2}{q_3^x} \cdot \frac{k_V[V]}{q_V K_{34}^x}\right) = \frac{k_2}{q_3^x} \cdot \frac{[V]}{K_V K_{34}^x} / N_{array}
\end{aligned}
\tag{22}
$$

The primary and compounded parameters in **Equation 22** are defined by Scheme one and Scheme 2, respectively, and are related to each other as shown in **Equation 3**. Here we have also defined the viomycin-binding constant as $K_V = q_V/[V]$. The differential equations corresponding to the scheme in **Equation 16** are:

$$
\begin{aligned}
\frac{dp_{2m}}{dt} &= -(q_2 + k_2)p_{2m} + q_{34}^x p_{34m} \\
\frac{dp_{34m}}{dt} \Big/ &= -(k_V^x[V] + q_{34}^x)p_{34m} + k_2 p_{2m} + q_V p_{4Vm} \\
\frac{dp_{4Vm}}{dt} \Big/ &= -q_V p_{4Vm} + k_V^x[V]p_{34m}
\end{aligned}
\tag{23}
$$

Integrating these equations with the initial conditions in **Equation 22** from zero to infinite time lead to the following set of algebraic equations for the average times $\tau_{4Vm}$, $\tau_{34m}$ and $\tau_{2m}$ the system spends in states $C_{4Vm}$, $C_{34m}$ and $C_{2m}$, respectively:

$$
\begin{aligned}
P_{2m} &= (q_2 + k_2)\tau_{2m} - q_{34}^x \tau_{34m} \\
P_{34m} &= (k_V^x[V] + q_{34}^x)\tau_{34m} - k_2 \tau_{2m} - q_V \tau_{4Vm} \\
P_{4Vm} &= q_V \tau_{4Vm} - k_V^x[V]\tau_{34m}
\end{aligned}
\tag{24}
$$

The general solution to the algebraic **Equation 25** is:

$$
\tau_2 = 1/q_2,
\tag{25}
$$

$$
\begin{aligned}
\tau_{34} &= \frac{1}{q_{34}^x}\left(P_{34}(0) + P_{4V}(0) + \frac{k_2}{q_2}\right), \\
\tau_{4V} &= \frac{1}{q_V}\left(P_{4V}(0) + k_V[V]\tau_{34}\right)
\end{aligned}
\tag{26}
$$

Writing the initial conditions in the elementary parameters of the scheme in **Equation 1** gives:

$$
\begin{aligned}
P_{4V}(0) &= \frac{[V]}{K_V} \frac{k_2 k_3}{q_3^x q_4^x} / N, \\
P_{34}(0) &= \frac{k_2}{q_3^x} \frac{k_3 + q_4^x}{q_4^x} / N, \\
P_2(0) &= 1/N
\end{aligned}
\tag{27}
$$

When chasing a cognate ternary complex we expect ribosomal states $C_{34}$ and $C_{4V}$ to dominate which leads to the following approximation:

$$
\tau_{diss}^c = \frac{k_3(k_2 + q_2)}{q_2 q_3^c q_4^c} + \frac{1}{q_V}\frac{[V]}{[V] + K_V} + \frac{[V]}{K_V}\frac{k_3(k_2 + q_2)}{q_2 q_3^c q_4^c}
\tag{28}
$$

The first term in **Equation 28** is the chase time in the absence of viomycin. The second term is the dissociation time for viomycin multiplied by the probability that viomycin is ribosome bound at the beginning of the chase. The third term reflects the prolonged chase time that is due to rebinding of viomycin that dissociates during the chase before dissociation of ternary complex. In near-cognate cases, we expect the back reactions in the schemes in **Equations 1 and 2** to be large, which leads to the following approximations for the average times in the different states:

$$\tau_{diss} = \tau_2 + \tau_{34} + \tau_{4V} \tag{29}$$

where

$$
\begin{aligned}
\tau_{20} &= \frac{1}{q_2}, \\
\tau_{34} &= \frac{1}{q_3^{nc}}\left(\frac{[V]}{K_I+[V]} + \frac{k_2}{q_2}\right) \\
\tau_{4V} &= \frac{1}{q_V}\left(\frac{[V]}{K_I+[V]} + k_V[V]\left[\frac{1}{q_3^{nc}}\left(\frac{[V]}{K_I+[V]} + \frac{k_2}{q_2}\right)\right]\right)
\end{aligned} \tag{30}
$$

and

$$K_I = \frac{K_V q_3^{nc} q_4^{nc}}{k_2 k_3} \tag{31}$$

The near-cognate chase experiment harbours several challenges. One is that dissociation of ternary complex from a viomycin-lacking ribosome is too fast to be measured with standard quench flow or stopped flow techniques. Another challenge is that the $K_I$-value in *equation 30* is much larger than the $K_V$-value due to large near-cognate backward rate constants.

We note that the ratio between the $K_I$-value in **Equation 10** above and $\tau_0$ in **Equation 26** gives

$$K_I \cdot \tau_0 \approx \frac{q_2}{(k_2+q_2)}\frac{q_3^{nc}q_4^{nc}}{k_Vk_3} \cdot \frac{1}{q_2}\frac{(k_2+q_2)k_3}{q_3^c q_4^c} = \frac{1}{k_V}\frac{q_3^{nc}q_4^{nc}}{q_3^c q_4^c} \tag{32}$$

It is clear that chase experiments in principle allows for estimations of $k_V$, $q_V$ and $\tau_0$ which allows for determination of the intrinsic selectivity $D = \frac{q_3^{nc}q_4^{nc}}{q_3^c q_4^c}$ in **Equation 12** from the $K_I$-value in **Equation 10** and $\tau_{diss}$ in **Equation 26**.

## Chase times and dissociation times in chase experiments

In experiments where a GTPase-deficient ternary complex, $T_{3m}$, is chased by a wild-type ternary complex, $T_3$, at a finite ratio between the free concentrations of $T_3$ and $T_{3m}$ the average chase time $\tau_{chase}$ is related to $\tau_{diss}$ by a factor $p_{chase}$. This factor is the probability that dissociation of $T_{3m}$ leads to a successful chase which in our experiments is signified by the peptidyl transfer reaction. Accordingly, $p_{chase}$ is given by

$$p_{chase} = \frac{[T_3](k_{cat}/K_m)}{[T_3](k_{cat}/K_m) + [T_{3m}](k_{cat}/K_m)_m} \tag{33}$$

Probability $p_{ch}$ connects the experimentally measured chase time, $\tau_{chexp}^c$, through the relation

$$\tau_{diss} = \tau_{chase} p_{chase} \tag{34}$$

