## [Decision Letter]

[Editors’ note: a previous version of this study was rejected after peer review, but the authors submitted for reconsideration. The first decision letter after peer review is shown below.]

Thank you for submitting your work entitled "Insights into the fidelity mechanism of mRNA decoding from characterization of viomycin induced miscoding in translation" for consideration by *eLife*. Your article has been reviewed by three peer reviewers, and the evaluation has been overseen by a Reviewing Editor, Alan Hinnebusch, and a Senior Editor. The reviewers have opted to remain anonymous.

Our decision has been reached after consultation between the reviewers. Based on these discussions and the individual reviews below, we regret to inform you that your work will not be considered further for publication in *eLife*.

Summary:

This paper examines the mechanism of miscoding in translation by the drug viomycin, concluding that drug binding increases initial selection of the near-cognate tRNA complexes rather GTP hydrolysis by the near-cognate ternary complex. Based on the results obtained and various theoretical considerations, the paper also challenges two aspects of the current induced-fit model for tRNA selection accuracy, concluding that (i) accuracy in the tRNA selection phase of decoding is enforced primarily by increased rates of near-cognate tRNA dissociation resulting from a higher rate of the back reaction from the codon recognition state to the initial binding state, rather than from a reduced rate of GTP hydrolysis for non-cognate versus cognate tRNAs; and (ii) interactions of bases A1492/A1493 with the tRNA anticodon stem loop is not restricted to cognate tRNA, with the unstated implication that these interactions play no important role in monitoring codon:anticodon helices in the A-site. Using pre-steady-state kinetic analysis with a quench-flow apparatus, they measure the catalytic efficiency (*k_cat_/K_m_*) for cognate versus near-cognate codons in the A-site in the presence of increasing concentrations of viomycin, for both GTP hydrolysis by the EF-Tu-GTP-aatRNAPhe ternary complex in the step of initial selection prior to proof-reading and for peptide-bond formation following proof-reading. The results show no effect of viomycin on *k_cat_ / K_m_* for cognate UUC, but a marked increase in *k_cat_ / K_m_* for near-cognate CUC; which leads to a large reduction in accuracy, defined as the ratio of (*k_cat_ / K_m_*)-cognate to (*k_cat_ / K_m_*)-near-cognate. Rather than finding that the near-cognate codon exhibits a much smaller *k_cat_ / K_m_*for peptide bond formation versus GTP hydrolysis owing to proof-reading, they are identical in the presence of drug, which means that proof-reading is disabled by the drug. They conclude, reasonably, that drug binding increases initial selection of the near-cognate complexes and that all near-cognates that pass initial selection go on to form peptide bonds.

The authors construct a kinetic model for drug action based on a current model in which the initial binding of cognate and near-cognate occurs at the same rate and only after transition to a codon-recognition state is the near-cognate tRNA rejected, returning to the initial binding state if GTP hydrolysis does not occur first. They propose that viomycin binds to the codon-recognition state and traps the tRNA in the A-site, preventing dissociation to the initial binding state and allowing GTP hydrolysis to occur. This has little effect on cognate tRNA because the rate of conversion to the initial binding state is slow and/or GTP hydrolysis is fast; but it dramatically increases the efficiency for near-cognate by preventing conversion to the initial binding state. In principle the drug could also increase the rate of GTP hydrolysis by near-cognate, but they approximate the equations that give a complete theoretical description of the kinetics with an expression in which the drug concentration required to increase the efficiency of each near-cognate reaction to one-half that of the cognate reaction, Ki, is governed by only the forward and back rates of the conversion from initial binding to codon-recognition states, and is independent of the rate of GTP hydrolysis. They go on to repeat the previous experiment for three different near-cognate codons and find a linear relationship between accuracy, (*k_cat_ / K_m_*)-cognate/(*k_cat_ / K_m_*)-near-cognate and *K_I_* for the four different near-cognates. They claim that this relationship indicates that the accuracy of initial selection for different near-cognates is determined primarily by different rates of tRNA rejection (the back reaction to initial binding state) rather than different rates of GTP hydrolysis in the manner proposed by the current model.

In the Discussion, they present the argument that, because viomycin binding to the A site requires that rRNA bases A1492/A1493 are in their "flipped out" conformation capable of interacting with the anticodon stem-loop of A site tRNA, that drug binding in the presence of a near-cognate codon requires that flipping out of the bases is not restricted to cognate codons, at odds with the recent model that these bases mediate an induced fit in the A site that triggers GTP hydrolysis. Thus, they propose that A1492/A1493 flipping out occurs dynamically when any tRNA is present in the A-site.

Summary of reviewer's comments:

There is agreement among the reviewers that your analysis of the mechanism of viomycin action has been well-designed and executed, and that it leads convincingly to the conclusion that drug binding increases initial selection of the near-cognate complexes, with attendant GTP hydrolysis, and that all near-cognates that pass initial selection go on to form peptide bonds owing to the fact that viomycin disables proofreading. In addition, it is considered significant that the inhibition of selection accuracy by the drug is in the millimolar range, whereas viomycin is known to inhibit tRNA translocation at low μM concentrations, implying that the main effect of viomycin on cellular translation is by inhibiting translocation and not by inducing decoding errors.

However, two of the reviewers object strongly to the second, more theoretical aspects of the paper where you argue from the linear relationship in Figure 3C that the accuracy of initial selection for different near-cognates is determined primarily by different rates of tRNA rejection rather than different rates of GTP hydrolysis-at odds with a leading model based on the work of Rodnina and colleagues. Both reviewers felt that this was too speculative, and that to argue convincingly against the prevailing model, it is necessary to directly measure ternary complex and/or aa-tRNA association/dissociation rates, in particular the dissociation rates of cognate and near-cognate tRNA, as you cannot currently determine whether viomycin slows down the dissociation of the ternary complex from the ribosome (as you conclude) or that viomycin accelerates GTP hydrolysis for near-cognates. It is difficult to follow the logic of your arguments, and it is questionable whether the approximation of the full set of equations required for a complete theoretical description of the kinetics you adopted in order to analyze the relationship between accuracy and *K_I_* is valid. The reviewing editor also had difficulty following these arguments and was concerned that the logic used to claim that accuracy is determined primarily by different rates of tRNA rejection rather than GTP hydrolysis may be circular, as you have used an approximation of the parameter *K_I_* in deriving equations 3-4 that doesn't include a term for GTP hydrolysis.

There was also objections to your second argument, that because viomycin binding requires A1492/A1493 to be in the flipped-out conformation, its binding to near-cognate/ribosome complexes necessitates that A1492/A1493 be flipped out for near-cognate as well as cognate tRNA-at odds with proposals that flipping out of A1492/A1493 is restricted to cognate tRNAs as a key step of the induced fit mechanism that accelerates GTP hydrolysis for cognate tRNAs. Your conclusion has been criticized because there is no crystal structure of ribosome/drug/near-cognate tRNA ternary complexes, and so the mode of viomycin binding in the presence of near-cognate tRNA is unknown; and also because there is no way to exclude that viomycin simply binds first and flips out the bases prior to tRNA binding. It was also noted in the discussion session that the available structural data do not assess the codon-recognition state, which is necessarily transient and distinct from the state probed by Ramakrishan and colleagues by stalling the ribosome with non-hydrolyzable GTP analogues or GTPase deficient mutants. Thus, we do not yet know the nature of the codon-anticodon interaction in the codon recognition state and viomycin's interactions with the ribosome and/or the codon-anticodon pair. Again, it was stated in the reviews that you should experimentally measure the conformational dynamics of A1492/A1493 during accommodation and viomycin binding to make mechanistic claims on this topic.

Finally, there was the criticism that extended into the discussion session among the reviewers, that you have failed to cite or discuss the previous study of Pape et al. (Rodnina) on the mechanistically similar antibiotic paromomycin, which involved separate measurements of the effect of paromomycin on GTP hydrolysis and aa-tRNA dissociation (measured using non-hydrolyzable GTP), leading to the conclusion that both steps are affected in the case of near-cognate aa-tRNA. It was objected that your study argues with that work without citing it, but with less experimental data (i.e. no tRNA dissociation analyses) to make the argument and using complex mathematical analyses based on questionable assumptions. While it was noted that there are reasons to question the conclusions of Pape et al., challenging them should be justified by the appropriate experimental data, such as measuring tRNA disassociation rates using a GTPase-deficient mutant of EF-Tu.

The reviewing editor originally recommended that the paper be sent out for review primarily because if your speculative arguments were judged to be compelling by expert reviewers, then they would have the impact of swaying the field towards accepting a revision of the tRNA selection mechanisms proposed by others in the field. However, with these speculations being strongly challenged by two of the three reviewers, it is now judged that the solid aspects of the paper concerning the mechanism of viomycin action and implications for its biological impact are not of sufficient general interest on their own for *eLife*.

*Reviewer #2:*

The manuscript by Holm et al. presents a clear kinetic description of the established miscoding impact of viomycin on tRNA selection (REF 5 and Wurmbach and Nierhaus Eur. J. Biochem 1983) using modern pre-steady state methods. A big picture message of the manuscript is that viomycin operates by preventing the ribosome from efficiently rejecting near-cognate tRNA at early stages of the selection process, which has larger implications regarding the fidelity mechanism. The authors' key conclusion from their finding is that viomycin impacts tRNA selection at a step prior to GTP hydrolysis and after conserved decoding site residues have contacted the mRNA-tRNA pair (i.e. during early steps of initial selection). This conclusion seems to be well supported by the data presented and fits with some, but not other, group's interpretations of the selection mechanism. Thus, the presented findings may have the broader impact of swaying the field towards accepting revision of the tRNA selection mechanisms proposed by Rodnina and colleagues that currently serves as the most widely accepted mechanistic framework. The experiments appear to have been rigorously performed, properly analyzed and kinetically modeled. The manuscript is well written and the findings appear to be appropriately interpreted in the context of existing literature. For these reasons, publication is recommended.

*Reviewer #3:*

Holm and Sanyal performed kinetic analysis of ribosome miscoding induced by the antibiotic viomycin. The authors measured the rate of GTP hydrolysis by EF-Tu and the rate of dipeptide formation to examine the effect of viomycin on initial tRNA selection and complete accommodation of tRNA to the A site, respectively. The authors found that accommodation of cognate tRNA is not affected by viomycin. By contrast, viomycin dramatically enhances rates of GTP hydrolysis and dipetide formation in the case of near-cognate tRNA. Kinetic data also suggest that viomycin completely abrogates proofreading step of tRNA selection. The experiments are well-executed. Observations made by the authors are of substantial interest to the protein synthesis community.

However, the authors went further and made a number of additional conclusions that are not related to experimental results in an obvious way. ("Viomycin binds to the ribosome during initial selection of tRNA, after binding of ternary complex but before GTP hydrolysis by EF-Tu". "In contrast to current ideas about 'induced-fit', accuracy in initial selection is achieved primarily by increase dissociation rates for near-cognate tRNAs rather than by decreased rates of GTP hydrolysis. Furthermore, the 'monitoring' bases A1492 and A1493 rapidly fluctuate between active and inactive conformations both when cognate and near-cognate tRNAs are present in the A site"). These conclusions are inferred from kinetic analysis of the authors' data. The authors acknowledge that they did not directly measure most of the rate constants required to obtain the complete kinetic description of tRNA accommodation. In particular, the authors did not directly measure the rate of tRNA disassociation of the codon recognition complex (the Rodnina's group used GTPase deficient mutant of EF-Tu to determine this rate constant experimentally). Hence, the authors' experiments cannot directly delineate whether viomycin slows down the dissociation of the EF-Tu ternary complex from the ribosome (as the authors conclude) or viomycin accelerates GTP hydrolysis. Furthermore, the authors did not directly examine the conformational dynamics of bases A1492 and A1493 of 16S rRNA during accommodation and viomycin binding. I could not completely follow the line of theoretical arguments that the authors used in support of their conclusions. I also could not fully understand the authors' justification for the "approximation" of complete kinetic description of the tRNA accommodation process to a simpler kinetic equation (equation 3) that the authors used. Hence, I might not be able to adequately judge the validity of main conclusions of the manuscript. Nevertheless, it seems to me that the authors' conclusions are made based on a number of assumptions that are not directly drawn from experimental results. Besides, the manuscript does not seem very accessible to a wide audience of readers and is rather addressed to a narrow group of scientists specialized in the kinetics of the tRNA accommodation process. I therefore doubt that this manuscript is suitable for publication in *eLife*.

*Reviewer #4:*

Holm and Sanyal employ pre-steady-state kinetics to study how the translation inhibitor viomycin affects the accuracy of aminoacyl-tRNA selection. The miscoding effect of viomycin has been reported before, e.g. by Marrero et al., 1980, also by Wurmbach and Nierhaus, 1982 (the authors do not cite the latter paper). The current study provides a more detailed assessment of miscoding of near-UUC codons by Phe-tRNAPhe. The authors find that the rates of GTP hydrolysis and dipeptide formation on near-cognate tRNA increase with the increasing concentrations of viomycin. The authors interpret that the concentration dependence reflects an equilibrium shift in the binding of near-cognate aminoacyl-tRNA to the A site. The authors find that the inhibition of selection accuracy (Ki) is in the millimolar range. This contrasts the effect of viomycin on tRNA translocation, which viomycin inhibits at low μM concentrations (Holms et al., 2016). As such, the study suggests that the main effect of viomycin on cellular translation is via the translocation step and not via decoding errors.

In addition to the conclusion about the mechanism of action of viomycin, the authors attempt to extend their findings to understand the mechanism of aa-tRNA decoding. Specifically, they ask whether the accuracy of selection is achieved via (1) "induced fit" of the ternary complex resulting in slow GTP hydrolysis on near-cognate aa-tRNA or (2) dissociation of near-cognate aa-tRNA. The authors also discuss the involvement of the decoding-center nucleotides A1492 and A1493 in tRNA selection. This part is speculative because the authors' findings do not directly report on tRNA association/dissociation or dynamics of the decoding center or even the binding mode (site and conformation) of viomycin in the presence of near-cognate tRNA. The rationalization of decoding accuracy in this work is not robust because it relies heavily on previous work, including crystal structures of near-cognate tRNAs or structures of viomycin-bound ribosomes. A mechanistic scheme is presented, in which viomycin binds following the ternary complex – but how was the order established, in which viomycin and the ternary complex bind? Surprisingly, the authors do not discuss a kinetic study of aa-tRNA miscoding by paromomycin (Pape, Wintermeyer and Rodnina, 2000), which addresses individual steps of ternary complex acceptance/rejection. That earlier work is particularly relevant because paromomycin' and viomycin's binding sites within h44 overlap and these antibiotics induce nearly identical rearrangements of A1492-A1493, so their mechanisms of tRNA miscoding are likely similar.

In summary, the detailed biochemical dissection of viomycin action on decoding in this work is interesting as it suggests that miscoding is not the primary mode of viomycin's antimicrobial action. However the implications for a general mechanism of translation accuracy are indirect and would require substantial additional work (e.g. direct measurements of ternary complex and/or aa-tRNA association/dissociation rates).

[Editors’ note: what now follows is the decision letter after the authors submitted for further consideration. After a subsequent appeal, the manuscript was accepted for publication.]

Thank you for submitting your work entitled "The mechanism of error induction by the antibiotic viomycin provides insight into the fidelity mechanism of translation" for consideration by *eLife*. Your article has been reviewed by two peer reviewers, and the evaluation has been overseen by a Reviewing Editor and a Senior Editor. The reviewers have opted to remain anonymous.

Our decision has been reached after consultation between the reviewers. Based on these discussions and the individual reviews below, we regret to inform you that your work will not be considered further for publication in *eLife*.

Although both reviewers felt that the work was well done, noting improvements over the original version of the paper, they also agreed that the findings add only incrementally to knowledge regarding the mechanisms of viomycin action and decoding in relation to previously published biochemical and structural studies on these topics. As such, neither felt that the paper satisfies the journal's standards for publishing work of the greatest importance to the field, and indicated that it would be of interest to only a very small group of specialists.

*Reviewer #1:*

Holm et al. performed kinetic analysis of ribosome miscoding induced by antibiotic viomycin. This manuscript is much improved in comparison to the earlier version of the paper. The manuscript is well written. Experiments and data analysis were meticulously done. The authors' conclusions are compelling. Important experiments with GTPase deficient mutant of EF-Tu that were missing from the previous version of the paper are now included. However, my enthusiasm is dampened because the manuscript does not offer many new insights regarding the mechanism of viomycin action or the mechanism of decoding. Most of authors' observations are consistent with previously published biochemical and structural studies of decoding/viomycin including authors' own works (e.g. Holm et al., 2016, Zhang et al., 2018). Furthermore, the authors conclude (and state this in the Abstract of manuscript) that viomycin-induced miscoding is not the cause of cell growth inhibition by viomycin. This conclusion further diminishes biological and medicinal significance of the study.

*Reviewer #2:*

The authors biochemically dissect how viomycin induces miscoding. They demonstrate that viomycin increases the rates of (1) EF-Tu-catalyzed GTP hydrolysis and (2) peptide bond formation for the near-cognate ternary complexes. Viomycin sensitivity (Ki) of different near-sense codons correlates with the accuracy for cognate tRNA over these codons (Figure 4), consistent with interference of viomycin with the decoding step(s). A competition assay is used to demonstrate that viomycin stabilizes near-cognate tRNA on the ribosome with catalytically-inactive EF-Tu, suggesting that the mechanism for GTPase activation on near-cognate tRNAs is due to stabilization of the ASL in the decoding center. These findings are consistent with previous biochemical and structural work showing that viomycin potently stabilizes tRNA in the A site, thus inducing miscoding and inhibiting translocation.

Although this work describes a well-designed and detailed biochemical study of viomycin's effect on near-cognate tRNA, these findings only incrementally add to what's been known. Increased binding of near-cognate tRNA to the A site in the presence of viomycin was demonstrated earlier (e.g. in Wurmbach and Nierhaus, Eur J. Biochem, 1983, but this work is not cited in the submitted manuscript). Next, the effect of viomycin on A1492 and A1493 constitutes bulk of the mechanistic insights discussed in the paper. But these insights are based almost entirely on previous structural studies because present biochemical assays do not directly report on the conformational changes in the decoding center. This discussion could as well be part of a review article, in the absence of the presented data. Furthermore, in vivo relevance of increased miscoding is not shown in this work. It is possible that these in vitro findings are not relevant because translocation inhibition is the prevalent mode of action of viomycin, so that mistranslated proteins do not contribute to cellular toxicity.

A minor note: The authors somewhat unexpectedly (for a broader readership) bring up aminoglycosides in the Introduction of the decoding mechanism. How relevant is this given that viomycin is not an aminoglycoside? If the modes of aminoglycosides and viomycin are deemed similar, what is the rationale for this study in the light of the known mechanism of miscoding by aminoglycosides?

---

## [Author Response]

[Editors’ note: the author responses to the first round of peer review follow.]

Summary of the review: Our manuscript concerns the mechanism of action of a ribosome targeting antibiotic drug, viomycin. In this work we show specifically how viomycin acts on selection of tRNA by the ribosome to induce missense errors in reading of the genetic code. We propose a complete mechanistic description of this mode of action of the drug and additionally, through comparison with our previous work on inhibition of translocation by viomycin, suggest that this is not the primary mode of action of the drug. This part of our manuscript was well received by the reviewers.

We then go further and show that the effects of viomycin on tRNA selection constrain the possible biochemical mechanisms through which the high accuracy of translation is achieved. We argue that our results contradict the prevailing ‘induced fit’ model. We also argue based on our results and on the published structures of the viomycin-bound ribosome that the so-called ‘monitoring bases’ A1492 and A1493 of the 16S rRNA are much more dynamic than has been proposed previously based on structural studies. These points raised in our manuscript were strongly criticized in the review.

What has been done: We have carried out a new set of experiments using a GTPase deficient mutant of EF-Tu to directly measure stabilization of ternary complexes on the ribosome by viomycin. These experiments address several concerns brought up in the review. We now show directly that viomycin strongly stabilizes both cognate and near/non cognate ternary complexes on the ribosome before GTP hydrolysis. We show that this stabilization is enough to completely explain the accuracy reducing effects of the drug. This provides direct experimental proof for the approximations made in our model.

A key argument in our manuscript concerns a correlation between the accuracy of a particular mismatched codon·anticodon pair during initial selection and the sensitivity of that pair to viomycin. We find that the more accurate a given mismatch is the more viomycin is required to yield a given misreading frequency for that mismatch. That is, the easier it is for the ribosome to reject a tRNA from the A site the harder it is for viomycin to bind to the ribosome while that tRNA is present. This propensity of viomycin to bind, while a tRNA is present in the ribosomal A site but before GTP hydrolysis by EF-Tu has occurred, is quantified in our manuscript by a single kinetic parameter, a *K_I_* value.

When estimating these *K_I_* values from our experimental data we rely on an approximation that was criticized in the review. We have updated the main text and that of the supplementary material to further clarify the nature of this approximation. To state it shortly we ignore the effect of the forward rate of GTP hydrolysis for near-cognate tRNA on the size of the *K_I_* value because it is so much smaller than the effective backwards rate of tRNA rejection. This assumption is true by definition for near/non cognate tRNA, it is also independent of the exact structure of the kinetic model used, if it were not true then the tRNA in question would be more likely to be accepted by the ribosome than to be rejected, and would by definition be a cognate tRNA.

Therefore, what we find is that the effective backwards rate of tRNA rejection, as measured by the *K_I_* value, and the accuracy of initial selection are strongly correlated. This is the evidence we rely on to strengthen our claim that the accuracy must be determined by tRNA rejection rates, rather than tRNA acceptance rates.

In our manuscript we also propose a hypothesis for the role of the monitoring bases A1492 and A1493 during tRNA selection. Since directly measuring the dynamics of these bases during tRNA selection on the ribosome is currently impossible by any experimental technique that we are aware of, we base our suggestion on our biochemical results and preexisting structures of viomycin-bound ribosomes.

We suggest that the conformation of the decoding center with A1492, A1493 and G530 interacting with the codon anticodon helix, is a necessary prerequisite for GTPase activation. To proceed to GTP hydrolysis any tRNA, be it cognate or non-cognate has to pass through this state. The propensity for a given tRNA to trigger GTP hydrolysis is then directly proportional to the lifetime of this state. The monitoring bases enhance the accuracy of tRNA selection by greatly decreasing the energy level of this state for cognate but not near/non-cognate tRNA. This same hypothesis has now been suggested by the Ramakrishnan lab (Shao et al., 2016) and demonstrated by Korestelev lab (Loveland et al., 2018). Our lab together with Joachim Frank’s lab also reached the same structural conclusion (Fislage et al., 2018). We do not claim to have directly observed the proposed behavior by the monitoring bases, but we do suggest that such a mechanism is compatible with our experimental data as well as all existing structural data on both viomycin-bound ribosomes and ribosomes with cognate or near-cognate tRNAs in the A site.

References:

Shao, S., Murray, J., Brown, A., Taunton, J., Ramakrishnan, V., and Hegde, R.S. (2016). Decoding Mammalian Ribosome-mRNA States by Translational GTPase Complexes. Cell 167, 1229-1240 e1215. 10.1016/j.cell.2016.10.046

Loveland, A.B., Demo, G., Grigorieff, N., and Korostelev, A.A. (2017). Ensemble cryo-EM elucidates the mechanism of translation fidelity. Nature 546, 113-117.

Fislage, M., Zhang, J., Brown, Z.P., Mandava, C.S., Sanyal, S., Ehrenberg, M., and Frank, J. (2018). Cryo-EM shows stages of initial codon selection on the ribosome by aa-tRNA in ternary complex with GTP and the GTPase-deficient EF-TuH84A. Nucleic Acids Res 46, 5861-5874.

[Editors' note: the author responses to the re-review follow.]

Although both reviewers felt that the work was well done, noting improvements over the original version of the paper, they also agreed that the findings add only incrementally to knowledge regarding the mechanisms of viomycin action and decoding in relation to previously published biochemical and structural studies on these topics. As such, neither felt that the paper satisfies the journal's standards for publishing work of the greatest importance to the field, and indicated that it would be of interest to only a very small group of specialists.Reviewer #1:Holm et al. performed kinetic analysis of ribosome miscoding induced by antibiotic viomycin. […]. This conclusion further diminishes biological and medicinal significance of the study.Reviewer #2:The authors biochemically dissect how viomycin induces miscoding. They demonstrate that viomycin increases the rates of (1) EF-Tu-catalyzed GTP hydrolysis and (2) peptide bond formation for the near-cognate ternary complexes. […] If the modes of aminoglycosides and viomycin are deemed similar, what is the rationale for this study in the light of the known mechanism of miscoding by aminoglycosides?

As you know, the current manuscript roots from an earlier manuscript submitted to *eLife* in 2016, with title “Insights into the fidelity mechanism of mRNA decoding from characterization of viomycin induced miscoding in translation”. In that manuscript, we presented with careful and precise biochemical work, how the antibiotic viomycin affects the accuracy of decoding. More importantly, we presented two vital insights about the fidelity mechanism of the decoding process on the ribosome, which were new to the field and not in line with views presented in the literature. These were, in brief:

i) The widely accepted ‘induced fit’ model for initial selection is incorrect; instead the accuracy in initial codon selection by ternary complex is decided by differential rejection rates of the cognate and near-cognate tRNAs before GTP hydrolysis by EF-Tu rather than by varied GTP hydrolysis rates for cognate and near-cognate tRNAs as proposed earlier.

ii) The monitoring bases in the decoding center, A1492 and A1493 flip out dynamically and interact with the codon anticodon helix irrespective of whether the tRNA is cognate or non/near cognate. We hypothesized that the propensity for a given tRNA to trigger GTP hydrolysis by EF-Tu and being accepted is directly proportional to the lifetime of this flipped-out state.

Two of the three referees and the reviewing editor reacted strongly against both of these findings (in 2016). The reviewing editor writes – ‘you find….that flipping out of the bases is not restricted to cognate codons, at odds with the recent model that these bases mediate an induced fit in the A site that triggers GTP hydrolysis’, and also, ‘its (viomycin’s) binding to near-cognate/ribosome complexes necessitates that A1492/A1493 be flipped out for near-cognate as well as cognate tRNA-at odds with proposals that flipping out of A1492/A1493 is restricted to cognate tRNAs as a key step of the induced fit mechanism that accelerates GTP hydrolysis for cognate tRNAs.’ The main complaint from one reviewer was that without a proper structure our hypothesis about the decoding mechanism could not stand. The reviewing editor criticized our conclusion because there was no crystal structure of ribosome/drug/near-cognate tRNA ternary complexes. We feel that two of the reviewers and the reviewing editor may have been preoccupied with the idea of the ‘induced fit’ model and were not open to results opposing it in the absence of high-resolution structures. There was however, no criticism concerning viomycin’s mode of action.

While we worked to gather more biochemical results to support our model of decoding, a paper presenting high-resolution structures of these very decoding states was published by the lab of Andrei Korostolev (Loveland et al., 2018). The results in this paper are fully in line with our proposed mechanism (already in 2016) that irrespective of the cognate or near/non cognate nature of the codon the monitoring bases are dynamically flipping in and out of h69. Now in 2019, when we submitted a more complete work presenting this mechanistic model for the fidelity of initial selection, supported by this recent structural data, our manuscript was rejected on the grounds that ‘the findings add only incrementally to knowledge regarding the mechanisms of viomycin action and decoding in relation to previously published biochemical and structural studies on these topics’.

We strongly object to the statement quoted in the paragraph above. We would like to point out that our state-of-the-art quantitative biochemical data are complementary to the recent structural data, not mutually exclusive. While the proposed mechanism of decoding can be speculated qualitatively from the now published structures, the quantitative aspects of our work go far beyond what can be deduced from just these structures. We were surprised that the reviewers and the reviewing editor minimized the importance of biochemical validation and extensions of the hypotheses proposed from structure, and imply that such biochemical experiments are useless.

We also question the referees’ reasoning based on comparison with our previous study on translocation inhibition by viomycin (Holm et al., 2016), that our manuscript is unimportant due to the demonstration that viomycin induced decoding error is not the main cause of the drug’s antimicrobial effect. We disagree with this argument. First, we feel both pathways need to be studied mechanistically to determine which of the two inhibition mechanisms is the strongest. Second, it has been demonstrated for other drugs (e.g. apramycin, an aminoglycoside), which cause both misreading and inhibition of translocation that the induction of decoding errors is correlated with the strength of the side effects observed in patients. Hence the demonstration of the fact that error-induction is not relevant to the antimicrobial activity of viomycin, a drug with severe side effects, is in our mind highly important for drug-development purposes.

Reviewer #2 is surprised that we mention aminoglycosides in the paper since the paper is about viomycin (which is not an aminoglycoside). What we have discussed is that in spite of the fact that these two classes of drugs share overlapping binding sites on the ribosome their modes of action are strikingly different, something that was not suggested from previous studies. In fact, the current opinion in the field is misleading. It is believed that as the drugs share a common binding site they must operate in the same way. Hence, we explicitly compare viomycin with aminoglycosides in our manuscript. Here the referee’s reasoning itself demonstrates why biochemical studies such as ours are an essential complement to structural studies as none of these questions could be addressed by the previous structural studies mentioned by both referees.

Reviewer #2 also suggests that our finding was already known from an article by Wurmbach and Nierhaus, Eur J. Biochem, 1983. We are aware of this paper, which presented a semi quantitative study on an incomplete translation system. We therefore feel it provides little mechanistic insight about viomycin’s actions on bacterial protein synthesis and chose not to discuss it.

This extended review and the editorial process has led us losing our priority in solving pertinent issues regarding fidelity mechanism of decoding on the ribosome. As both the past and the present rejections appear to be based on bias from the reviewers, we feel obliged to appeal the editorial decision and ask for a new review of the current manuscript.